# Understanding the sulphur-oxygen exchange process of metal sulphides prior to oxygen evolution reaction

Yang Hu [1,2,7], Yao Zheng [3,7], Jing Jin [1,7], Yantao Wang [1], Yong Peng[4,5] ✉,
Jie Yin [1,2], Wei Shen [1], Yichao Hou [1], Liu Zhu [4,5], Li An[1,2], Min Lu [1,2],
Pinxian Xi [1,2] ✉ & Chun-Hua Yan[1,2,6]

Dynamic reconstruction of metal sulphides during electrocatalytic oxygen evolution reaction (OER) has hampered the acquisition of legible evidence for comprehensively understanding the phase-transition mechanism and electrocatalytic activity origin. Herein, modelling on a series of cobalt-nickel bimetallic sulphides, we for the first time establish an explicit and comprehensive picture of their dynamic phase evaluation pathway at the pre-catalytic stage before OER process. By utilizing the in-situ electrochemical transmission electron microscopy and electron energy loss spectroscopy, the lattice sulphur atoms of $(NiCo)S_{1.33}$ particles are revealed to be partially substituted by oxygen from electrolyte to form a lattice oxygen-sulphur coexisting shell surface before the generation of reconstituted active species. Such S-O exchange process is benefitted from the subtle modulation of metal-sulphur coordination form caused by the specific Ni and Co occupation. This unique oxygen-substitution behaviour produces an $(NiCo)O_xS_{1.33-x}$ surface to reduce the energy barrier of surface reconstruction for converting sulphides into active oxy/hydroxide derivative, therefore significantly increasing the proportion of lattice oxygen-mediated mechanism compared to the pure sulphide surface. We anticipate this direct observation can provide an explicit picture of catalysts' structural and compositional evolution during the electrocatalytic process.

Oxygen evolution reaction (OER) has a complicated mechanism and kinetics at the anode of a water electrolyser due to the four-electron transfer process[1,2]. Over the past decades, many kinds of non-precious metal catalysts, such as transition metal (TM) oxides, chalcogenides, hydroxides etc., have been developed to be the benchmark electrocatalysts[3–11]. Among them, dual-transition metal sulphides

exhibit improved electroconductivity and OER activity under alkaline conditions by taking advantage of the bimetal synergistic effect[12–18]. In practice, TM-based electrocatalysts are revealed to go through a surface reconstruction process to form oxides/hydroxides shell active sites under the action of electrolyte hydrolysis and anode polarization[19,20]. Nowadays, the phase-transition process, identification

[1]State Key Laboratory of Applied Organic Chemistry, College of Chemistry and Chemical Engineering, Lanzhou University, Lanzhou 730000, China. [2]Frontiers Science Center for Rare Isotopes, Lanzhou University, Lanzhou 730000, China. [3]School of Chemical Engineering and Advanced Materials, The University of Adelaide, Adelaide, SA 5005, Australia. [4]School of Materials and Energy, Lanzhou University, Lanzhou 730000, China. [5]Electron Microscopy Centre, Lanzhou University, Lanzhou 730000, China. [6]Beijing National Laboratory for Molecular Sciences, State Key Laboratory of Rare Earth Materials Chemistry and Applications, PKU-HKU Joint Laboratory in Rare Earth Materials and Bioinorganic Chemistry, College of Chemistry and Molecular Engineering, Peking University, Beijing 100871, China. [7]These authors contributed equally: Yang Hu, Yao Zheng, Jing Jin. ✉e-mail: pengy@lzu.edu.cn; xipx@lzu.edu.cn

of metallic active sites and corresponding OER mechanism of TM oxides has been comprehensively investigated[21]. However, the structural evolution of TM sulphides and their activity origin are poorly known, especially for the dynamic reconstruction induced by the OH ions in the electrolyte and S atom in the lattice of electrocatalyst[22]. Therefore, revealing the phase reconstruction mechanism of TM sulphides during the OER process is critical.

By using ex-situ techniques such as transmission electron microscopes (TEM) characterisation, the structural and elemental transition of Ni-Fe disulphide to hydroxide process has been revealed after the OER process, which was further oxidised as the crystalline Ni-Fe oxyhydroxide shell to promote the electrocatalytic reaction[23]. Ordinary Raman spectroscopy was also used to reveal the formation of the $Ni(OH)_2$/NiOOH active phases on the $Ni_3S_2$ during OER[24]. Some studies showcase a partial oxidation mechanism to form a sulphide/oxide core/shell structure, while another viewpoint suggests that the unstable NiS and CoS was completely oxidised into their corresponding oxides under alkaline condition, which is supported by the X-ray photoelectron spectroscopy (XPS) technology[25,26]. Recently, it has been proposed that the Nickel chalcogenides actually form a mixed anionic (hydroxy)chalcogenide surface for boosting water oxidation, where the metal is coordinated to both anionic and OH[27]. Due to the lack of visible and direct evidence on the new-formed species, it is difficult to reach an agreement on the reconstruction process that contains some nonequilibrium states, which actually drive the formation of different active species during the reaction process.

With the rapid development of in-situ characterisation technology, real-time monitoring of the chemical and physical evolution of the catalyst surface during electrocatalytic reactions has become a reality. The operando Raman and X-ray absorption spectroscopy can help us to reveal the detailed structure and composition evolution of multivalent TM sulphides during the water decomposition process[28–30]. Nevertheless, the monitoring results of these spectra reflect the average information about the milligram-level powders from a macro perspective; the local evolution pathway of crystal structure and element valence state cannot be directly visualised. By taking advantage of an in-situ electrochemical TEM (in-situ EC-TEM) methodology, the formation of a nanometric cobalt (oxyhydr)oxide-like phase on the surface of $Co_3O_4$ nanoparticles during OER can be directly observed[31]. Besides, a suite of correlative operando scanning probes and X-ray microscopy techniques are utilised to monitor the generation of the $\alpha$-$CoO_2H_{1.5}$·$0.5H_2O$ on single-crystalline $\beta$-$Co(OH)_2$ platelet particles and the valance state variation of metal ions at pre-catalytic voltages[32]. Although these visual in-situ characterisation techniques realise the observation of the local evolution of catalysts, it mainly focuses on holistic structural remodelling and neglects the front path that drives the reconstruction process. More importantly, due to the widely observed sulphur atom loss during the sulphide's transform into a corresponding active oxide/hydroxide electrocatalyst, the TM sulphides showcase a more complicated restructuring mechanism compared to the pure oxides and hydroxides.

Herein, utilising a suite of in-situ spectroscopy and microscopy techniques, we, for the first time, report a full and detailed dynamic phase evaluation of NiCo-based sulphides at the pre-catalytic stage before the generation of reconstituted active species under alkaline OER operation. Driven by the pre-catalytic voltage and hydroxyl molecules in solution, the lattice sulphur atoms on the surface of $(NiCo)S_{1.33}$ particles are revealed to be partially substituted by oxygen and further induce the formation of a lattice oxygen-sulphur coexisting surface. This kind of newly formed oxygenated surface can reduce the energy barrier for superficially reconstructing into the NiCo-oxy/

hydroxide as the active electrocatalyst to further boost the OER process, which is very different to the conventional observation in NiCo-oxides and hydroxides. This additional and unique substitution process of $(NiCo)S_{1.33}$ helps to promote surface reconstruction and achieve a considerable enhancement of OER performance compared to the other NiCo-based sulphides. This work showcases visual and direct evidence for elaborating the dynamic evolution pathway of bimetallic sulphides at the pre-catalytic stage and linking the internal mechanism of surface remodelling with the atomic occupation forms of metal atoms.

## Results and discussion
### Atomic occupation regulation
TM sulphides have various crystal structures determined by the specific metal-sulfur coordination form. According to the phase-dependent synthesis strategy (Supplementary Fig. 1), four cobalt-nickel bimetallic sulphides with different compositions (e.g. $(NiCo)S_{0.89}$, $(NiCo)S$, $(NiCo)S_{1.33}$ and $(NiCo)S_2$) were obtained. The powder X-ray diffraction (XRD) characteristic peaks of as-prepared samples match with that of four typical sulphide structures with the space group of Fm-3m, P63/mmc, Fd-3m and P3-a, respectively (Fig. 1a and Supplementary Fig. 2). According to the Fourier transform (FT) K-edge X-ray absorption fine structure (EXAFS) of $(NiCo)S_{1.33}$ in Fig. 1b, two separate peaks (peaks II and III) of Co at -2.6–3.1 Å can be assigned to the features of $Co_{Oh}$ (octahedral site) and $Co_{Td}$ (tetrahedral site), while Ni only occupies the octahedral sites[33]. In contrast, the other three bimetallic sulphides revealed the random occupation of metal atoms, in which Ni and Co have the same probability of occupying octahedral or tetrahedral sites (Supplementary Figs. 3, 4). In addition, two intensity maxima at -1.6 and 2.6 Å can be observed in the EXAFS wavelet transform images of both Co and Ni in $(NiCo)S_{1.33}$ (Supplementary Fig. 5), which were assigned to the Co-S and Ni-S, $Co_{Oh}$/$Co_{Td}$ and $Ni_{Oh}$ contributions. Spherical aberration-corrected transmission electron microscopy (Cs-STEM) was employed to clearly verify the crystalline differences of these four nickel-cobalt bimetallic sulphides. As shown in Fig. 1c and Supplementary Fig. 6, the $(NiCo)S_{1.33}$ exhibited a typical antispinel structure with the crystal orientation of [011]. The $Ni_{Oh}$ and $Co_{Oh}$/$Co_{Td}$ occupations and different coordination forms of Co-S and Ni-S could be directly visualised by the atomic STEM-energy dispersive X-ray (STEM-EDX) elemental mapping (Fig. 1d and Supplementary Fig. 7), which is consistent with the EXAFS results. Meanwhile, the crystal structure and elemental distribution characterisation of $(NiCo)S_{0.89}$, $(NiCo)S$ and $(NiCo)S_2$ revealed that the Co and Ni atoms inside each structure have a similar coordination form with S atoms due to the same occupation of two metal atoms (Supplementary Figs. 8–10). And the elemental mapping of Co and Ni at low magnification further proved their uniform distribution in each particle of $(NiCo)S_{0.89}$, $(NiCo)S$ and $(NiCo)S_2$ (Supplementary Fig. 11).

The $L_3$-edges electron energy loss spectroscopy (EELS) revealed that the Co and Ni $L_3$-edge of $(NiCo)S_{1.33}$ was significantly shifted to the high-energy direction compared to the other three sulphides (Fig. 1e and Supplementary Figs. 12–16). Referring to the Co and Ni K-edge EXAFS spectra (Supplementary Fig. 17), these results suggest an enhancement of metal-sulfur bond covalency due to the regulation of the unique M-S coordination form. Furthermore, the d-band centre ($\varepsilon_d$) of the Co and Ni atoms in these sulphides was calculated to estimate the position of an average value for Co and Ni d orbitals. The metal d-band centre of $(NiCo)S_{1.33}$ is calculated to be closer to the Fermi level (Fig. 1f, Supplementary Figs. 18–20 and Supplementary Table 1), implying the more electronic states near the Fermi level and stronger adsorption capacity for reaction intermediates than the other three sulphides. In detail, the Co d-band centre of these sulphides were revealed to be closer to the Fermi level than those of Ni, which suggested that Co should be a dominant metal active site for adsorbing the reaction intermediates during the OER process. Combining the

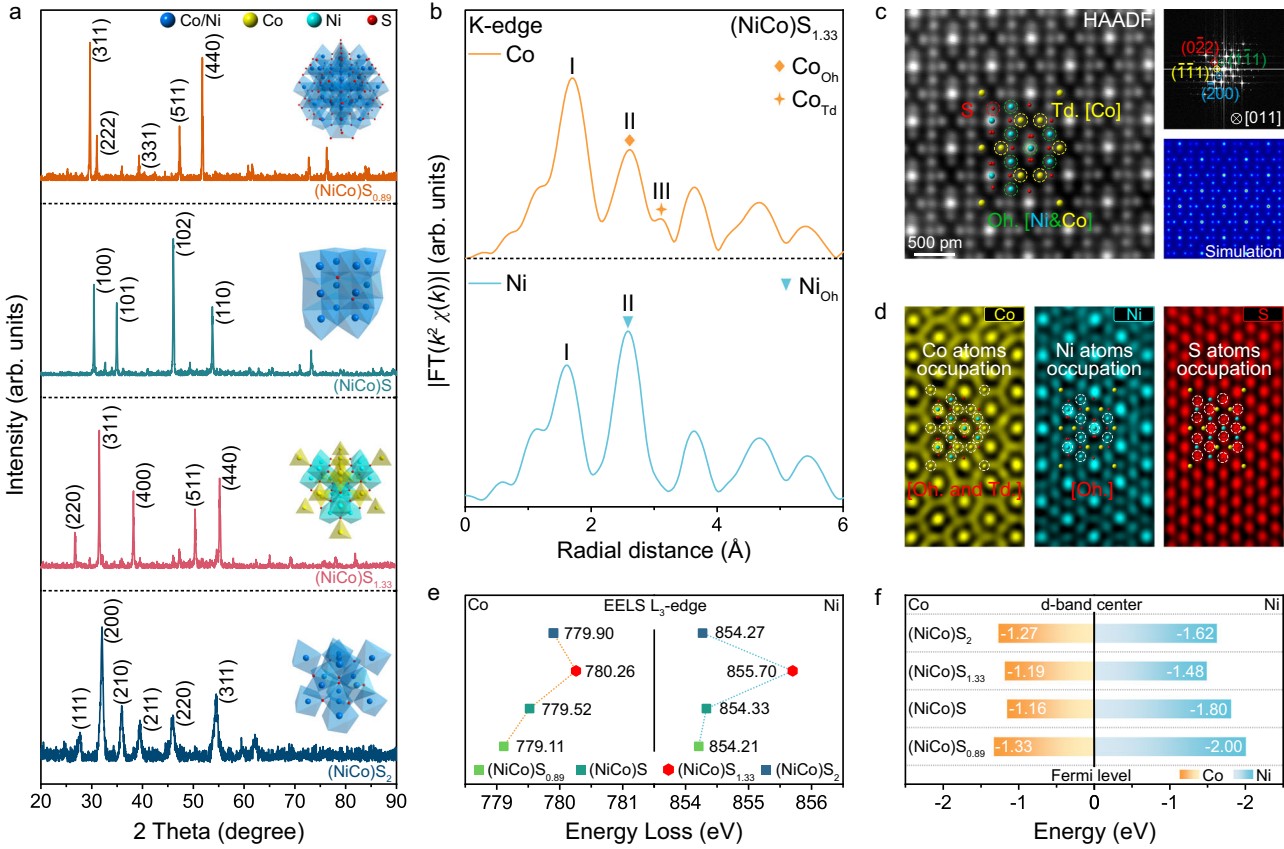

**Fig. 1 | Structural and chemical characterisation of as-prepared nickel-cobalt bimetallic sulphides. a** Powder XRD patterns of four synthesised nickel-cobalt bimetallic sulphides. Inset, the crystal structure diagram of $(NiCo)S_{0.89}$, $(NiCo)S$, $(NiCo)S_{1.33}$ and $(NiCo)S_2$. **b** FT K-edge EXAFS spectra of $(NiCo)S_{1.33}$. **c** Atomic STEM-HAADF image of $(NiCo)S_{1.33}$ with corresponding FFT and HAADF simulation of $(NiCo)S_{1.33}$ along the [011] orientation. $Co_{Td}$ occupations are labelled with yellow circles. $Co_{Oh}$ and $Ni_{Oh}$ occupations are labelled with green circles. The occupation of S atoms are labelled with red circles. **d** Atomic EDX elemental mapping of Ni, Co and S. $Co_{Td}$ and $Co_{Oh}$ occupations are labelled with white circles. $Ni_{Oh}$ occupations are labelled with white circles. The occupation of S atoms are labelled with white circles. **e** Peak position of Co and Ni $L_3$-edge collected from the EELS spectra of $(NiCo)S_{0.89}$, $(NiCo)S$, $(NiCo)S_{1.33}$ and $(NiCo)S_2$ L-edge. **f** Co and Ni d-band centre of $(NiCo)S_{0.89}$, $(NiCo)S$, $(NiCo)S_{1.33}$ and $(NiCo)S_2$.

analysis of structural characterisation, EELS spectra and density functional theory (DFT) calculations, we could confirm that the specialisation of the metal-sulfur coordination form, induced by the change of metal atom occupation, has caused the electronic environment of the metal atoms in $(NiCo)S_{1.33}$ to be different from the other three sulphides.

## Real-time observation of the pre-catalytic process

Compared to the other three samples with random occupation, the $(NiCo)S_{1.33}$ presented a better OER performance with an overpotential of 302 mV at a current density of 10 mA/cm² (Fig. 2a and Supplementary Figs. 21, 22). More importantly, the obvious pre-oxidation behaviour of $(NiCo)S_{1.33}$ has been observed at the pre-catalytic stage before the OER process in linear sweep voltammetry (LSV) and Cyclic Voltammetry (CV) (Supplementary Fig. 23), and the catalytic performance of $(NiCo)S_{1.33}$ exhibited a significant improvement after five CV cycles, while the other sulphides showing a decreasing trend (Supplementary Fig. 24). These results imply a different reaction pathway of $(NiCo)S_{1.33}$ at the pre-catalytic stage, which could be considered as an intrinsic cause of different catalytic performance.

In order to reveal the local structural and composition transformation of the material in the pre-catalytic stage, in-situ electron microscopy was employed to directly observe the surface change of $(NiCo)S_{1.33}$. As shown in Fig. 2b, the electrochemical chip with the as-prepared $(NiCo)S_{1.33}$ attached to the working electrode was loaded into the in-situ liquid sample holder, and passed 0.1 M KOH as the

electrolyte (see Method for details). The oxidation peak under the in-situ condition exhibited similar behaviour as the ex-situ CV curve (Supplementary Fig. 25), which indicates that the microscopic phenomena observed by in-situ electron microscopy during the pre-catalytic stage can directly reflect the real evolution of the catalyst in the macroscopic test. And the in-situ CV curve also revealed that the potential range of the pre-catalytic process is between 0.4 to 0.9 V (vs Pt pseudo-Reference). More importantly, there is no significant difference between the CV curves measured under the condition of continuous electron beam irradiation and when the electron beam is turned off (Supplementary Fig. 26). Besides, the current density did not exhibit significant fluctuation when the electron beam switched from the off state to on state. And the current density kept a flat trend during the 5 minutes of electron beam irradiation (Supplementary Fig. 27). Thus, it could be confirmed that the electron beam does not have a significant impact on the structural evolution of the catalyst. According to the in-situ CV curve, we applied the constant voltage of 0.9 V for ensuring that the catalyst undergoes a complete pre-catalytic process to obtain a full picture of the pre-catalytic stage (Fig. 2c). As shown in Fig. 2d, e, after 21 s as the voltage was applied, a new shell gradually appeared on the surface of the particles and the diameter of the particles increased by 14.0 nm, which was induced by the hydroxide intercalation. Under the continuous application of voltage, the reconstruction rapidly intensified (Supplementary Movie 1). The average thickness of the hydroxide shell increased from 27.75 nm in two seconds (24–26 s), and the overall diameter increased to 597.3 nm.

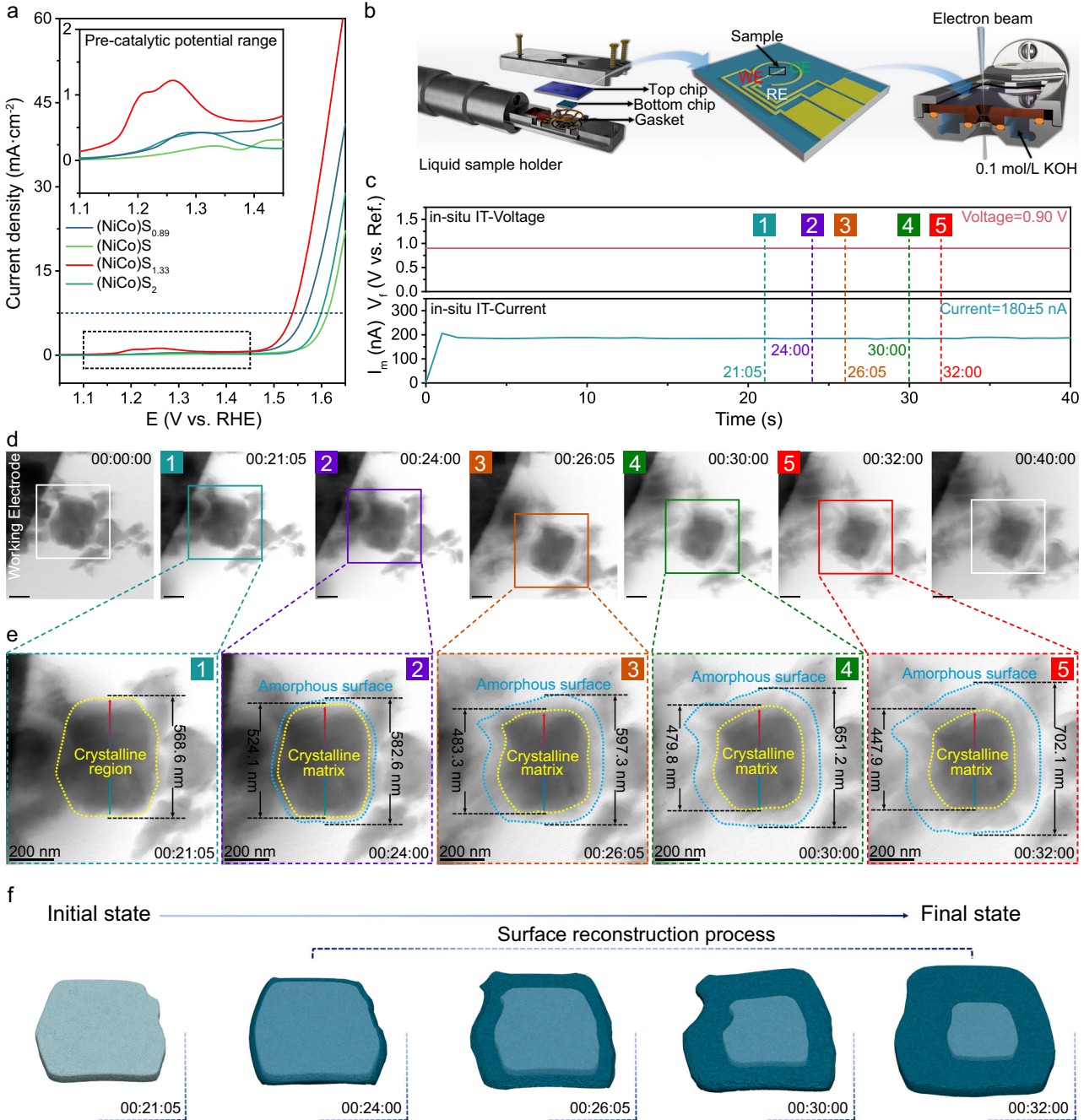

**Fig. 2 | Real-time observation of the surface reconstruction process. a** iR-corrected LSV at a scan rate of 5 mV s⁻¹. **b** Schematic diagram of in-situ TEM experiment. WE (red), CE (green) and RE (white) represent the working electrode, a counter electrode and a reference electrode, respectively. **c** The applied potential and corresponding current during the in-situ IT test. **d**–**f** The surface reconstruction under the applied voltage of 0.9 V and corresponding schematic diagram of (NiCo)S₁.₃₃. Scale bar = 200 nm.

It should be noted that the diameter of the crystal core reduced with the reconstruction processing, indicating that the process of surface reconstruction would continue to diffuse into the particle. For this central particle, the diffusion depth of the surface reconstruction is ~69.35 nm. After the shell wrapped on the surface with a certain thickness of 127.1 nm, the reconstruction process was terminated and the particle surface reached a stable state (Supplementary Fig. 28 and Supplementary Movie 1). In addition, in order to exclude the Oswald ripening process in this electrochemical environment, several control experiments were conducted. First of all, the morphology and size of a (NiCo)S₁.₃₃ nanoparticle, which has the comparable size as the central particle in Fig. 2e, without contacting the working electrode was

continuously monitored under the applied voltage of 0.9 V (Supplementary Fig. 29a). It could be found that the morphology, size, and surface structure of this nanoparticle did not change during a period of up to 40 s (Supplementary Fig. 29b). And according to the dissolution mechanism of small particles during the Oswald ripening, a small particle with a diameter of 46.35 nm was selected to observe the diameter variation under the applied voltage of 0.9 V (Supplementary Fig. 30a). As shown in Supplementary Fig. 30b, the diameter of this small particle increased to 71.23 nm within 30 s, which suggested that there is no dissolution process occurred in the electrochemical environment. At last, the observation of the surface reconstruction process of an individual (NiCo)S₁.₃₃ particle under the applied voltage

of 0.9 V could provide a more convincing result (Supplementary Fig. 31), which exhibited similar behaviour as those particles in Fig. 2d during the surface reconstruction process. Based on the above results, we could basically exclude the Ostwald ripening process in this electrochemical environment, which further proved that the diameter increase and the morphological variation of the $(NiCo)S_{1.33}$ nanoparticles in Fig. 2e should be attributed to the surface reconstruction process. Notably, the morphology of $(NiCo)S_{1.33}$ surface did not undergo a significant change and maintained a good crystallinity in 21 s with the applied voltage of 0.9 V, which indicates that none of the hydroxide ion intercalations occurred on the catalytic surface during this time period (Fig. 2f).

### The exchange of lattice sulfur with oxygen

We further reveal a more detailed insight into the variation that was taken place on the surface of the material during the initial 21 s (Supplementary Fig. 32). We exchanged another chip with a fresh sample of $(NiCo)S_{1.33}$ and applied a series potential of 0, 0.9, 1.0, 1.1, 1.2 and 1.3 V (vs Pt pseudo-Reference) for 20 s, respectively. As shown in Fig. 3a, the degree of surface reconstruction continues to deepen with the voltage increasing. As the applied potential is higher than 0.9 V, the surface of the particle began to reconstruct and gradually transforms into amorphous metal oxyhydroxide. According to the in-situ elemental mapping of S and O (Fig. 3b), it could be observed that the elemental content of S on the $(NiCo)S_{1.33}$ surface dropped sharply during the voltage increasing from 0.9 to 1.0 V. Interestingly, the average O-content under 0.9 V exhibited a significant promotion compared to

OCV (Supplementary Fig. 33 and Supplementary Table 2). Moreover, the mixed elemental map (Fig. 3c) and their corresponding extraction of the elemental distribution curve (Fig. 3d) revealed that the oxygen and sulphur atoms are coexisted in the range of 8.0 nm at the surface with comparable content. It is worth noting that the surface of $(NiCo)S_{1.33}$ particle maintained the crystal structure without any reconstruction behaviour at this stage, which exhibited a uniform crystal structure as the bulk phase. In contrast, significant surface reconstruction of the counterpart $(NiCo)S_{0.89}$ could not be observed until the applied potential was higher than 1.1 V (Supplementary Fig. 34a). Meanwhile, during the increase of voltage up to 1.1 V, there was no significant loss of sulfur on the surface of $(NiCo)S_{0.89}$ and the O-content exhibited a slow ascent process (Supplementary Fig. 34b, c and Supplementary Table 3). $(NiCo)S$ and $(NiCo)S_2$ also showcased the same variational tendency as $(NiCo)S_{0.89}$ (Supplementary Figs. 35, 36 and Supplementary Table 3). Therefore, it could be suggested that the lattice sulphur atoms were partially substituted by the oxygen from the hydroxide ion in an electrolyte to form the S-O coexisting phase in the sulphide lattice at the pre-catalytic stage, which was not visually verified by the conventional spectrum technology and very different with the other three counterparts. In addition, in order to exclude the irradiation effect of an electron beam on the leaching of sulphur, several control experiments were performed under different conditions. First of all, the irradiation experiment with a different irradiation intensity of electron beam was carried out respectively in vacuum and liquid environments to investigate whether electron beam irradiation would cause the leaching of sulphur. As shown in Supplementary Figs. 37–40, the

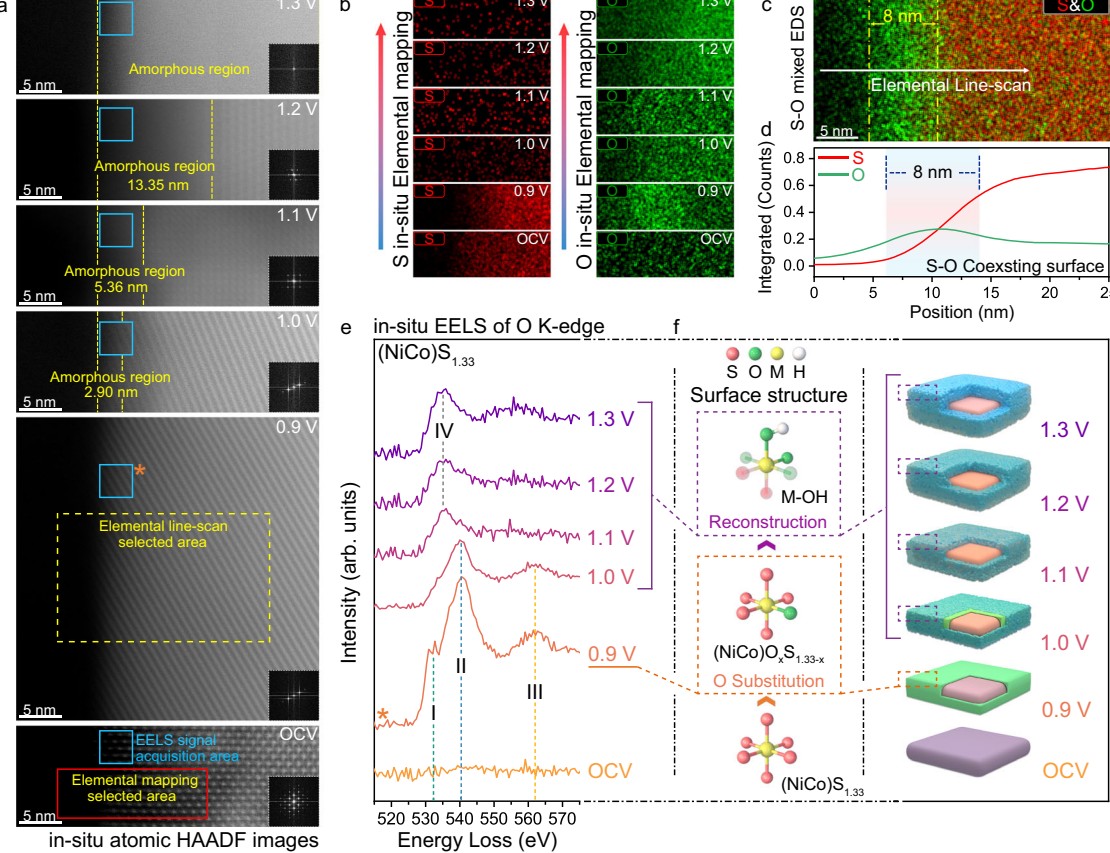

**Fig. 3 | The exchange of lattice sulphur with oxygen. a** In-situ atomic HAADF images of $(NiCo)S_{1.33}$ after constantly applying different potentials for 20 s. **b** In-situ elemental mapping of S and O under different applied potentials. **c** S-O mixed elemental mapping of $(NiCo)S_{1.33}$ after constantly applying potentials of 0.9 V for 20 s. **d** The corresponding elemental distribution curve extracted from (**c**). **e** In-situ EELS spectra of O K-edge under different applied potential. The I, II and III peaks can be assigned to the hybridisation of O $2p$ with Co $3d$, Ni $3d$ and Co $4sp$ orbitals, respectively. The IV peaks can be assigned to the characteristic peaks of oxygen in an amorphous structure. **f** Schematic diagram of lattice sulfur-oxygen substitution process.

relative content and distribution of S did not change significantly after 10 min irradiation at the screen current of 30 and 300 pA. This result revealed that continuous electron beam irradiation is not the cause of sulphur leaching in these sulphides. Meanwhile, the in-situ TEM heating experiment and Thermogravimetric analysis (TGA) were conducted to investigate whether the high temperature around the catalyst particles, which may cause by the irradiation effect of an electron beam, will lead to the leaching of sulphur in $(NiCo)S_{1.33}$. The relative content and distribution of S were not observed to change significantly after heating up to 500 °C (Supplementary Figs. 41, 42). And the TGA result has also shown that the overall quality of $(NiCo)S_{1.33}$ powder did not reduce significantly during the process of heating up to 500 °C in the nitrogen environment (Supplementary Fig. 43). Besides, the electrolyte at room temperature was constantly flowing during the in-situ EC-TEM experiment, which would counteract the heating effect caused by electron beam irradiation at a certain extent and keep the catalyst particles in a constant temperature state. Thus, it could be confirmed that the leaching of sulphur observed in the in-situ EC-TEM experiment is due to the progression of the electrochemical reaction.

To further expound the lattice sulfur-oxygen exchange process, we conducted a more detailed analysis of the atomic electronic environment of oxygen by employing EELS under a series of applied potentials to reveal the evolution of the ionic bond between metal and oxygen. The in-situ EELS of O K-edge (Fig. 3e) under 0.9 V showcased three peaks, i.e. I, II and III, that can be assigned to the hybridisation of O $2p$ with Co $3d$[34,35], Ni $3d$[36,37] and Co $4sp$[38,39] orbitals, respectively. This result could prove that oxygen has entered into the surface lattice of $(NiCo)S_{1.33}$ by substituting a part of sulphur atoms and formed ionic bonds with both Co and Ni atoms (Fig. 3f), while the whole catalyst remained the crystal structure as sulphide matrix. As the surface reconstruction process proceeds, the dissociation of surface crystal structure induced the bond breaking of Co-O and Ni-O, which manifested as the evolution of characteristic peaks. Combining the above results, we can determine that under the potential of 0.9 V, the lattice sulfur atoms in the $(NiCo)S_{1.33}$ surface are exchanged by oxygen atoms, which further induces the formation of oxygen-sulphur coexisting surface as $(NiCo)O_xS_{1.33-x}$. Whilst, the lattice S in $(NiCo)S_{0.89}$, $(NiCo)S$ and $(NiCo)S_2$ surface was not substituted by O under the pre-catalytic voltage of 1.1, 1.2 and 1.1 V, respectively, to form the ionic bonds with metal atoms (Supplementary Fig. 44). Moreover, in order to confirm the threshold voltage of oxygen-substitution process in $(NiCo)S_{1.33}$, the changes of the crystal structure, elemental distribution and metal-oxygen bonds on the surface under lower potential (0.5, 0.6, 0.7 and 0.8 V) were captured. According to the atomic HAADF images, elemental mapping and corresponding elemental distribution curves (Supplementary Fig. 45a–c), it could be found that the $(NiCo)S_{1.33}$ surface still maintained good crystallinity and no enrichment of oxygen element was detected under the lower potential conditions after 20 s. Combined with O-EELS (Supplementary Fig. 45d), it could be confirmed that there is no generation of metal-oxygen bond on the $(NiCo)S_{1.33}$ surface under the lower potentials. Besides, the variation of $(NiCo)S_{1.33}$ surface has been further investigated after applying the voltage of 0.9 V for 10 seconds. As shown in Supplementary Fig. 46a, b, the $(NiCo)S_{1.33}$ surface had a certain degree of oxygen enrichment with a uniform crystal structure, and the width of the oxygen enrichment area is 3.68 nm (Supplementary Fig. 46c, d). Combined with O-EELS (Supplementary Fig. 46e) and the change of sulphur-oxygen content under different voltages (Supplementary Fig. 47 and Supplementary Table 4), it can be confirmed that M-O bonds were formed on the surface. This observation under the applied voltage of 0.9 V after 10 s is consistent with that observed after 20 s (Fig. 3c, d), except for the difference in the thickness of the sulphur-oxygen coexistence surface. Based on these observations, it could be determined that the sulfur-oxygen substitution process will start until the voltage raises up to 0.9 V at the pre-catalytic stage and the thickness of $(NiCo)O_xS_{1.33-x}$ shell thickens with the increase of sulfur-oxygen substitution time. Whilst, none of the surface oxidation will occur when the applied voltage is lower than 0.9 V. Therefore, we suggest that the threshold voltage of the lattice sulfur-oxygen substitution process occurred in $(NiCo)S_{1.33}$ surface is 0.9 V under the in-situ condition. These results have revealed a distinct per-oxidation process of $(NiCo)S_{1.33}$ as the entrance of oxygen atoms into the crystal lattice to form the $(NiCo)O_xS_{1.33-x}$ surface at the pre-catalytic potential of 0.9 V before fully reconstructing into active metal-oxy/hydroxide. This oxygen-substitution of surface lattice sulfur can effectively reduce the threshold voltage of surface reconstruction, thereby reducing the overpotential of the bimetallic sulphide and promoting the occurrence of OER.

## Origin and influence of the lattice sulfur-oxygen substitution process

In order to investigate the internal reason for this unique S-O substitution process occurred in $(NiCo)S_{1.33}$ among these sulphides, we analysed their intrinsic electronic property differences in-depth by the DFT calculation (see method for details). It showed that the p-band centre of S in $(NiCo)S_{1.33}$ is closest to the Fermi level among these as-prepared sulphides, indicating a higher adsorption capacity to binding the dissociative hydroxide ions in electrolyte for further substituting with oxygen (Fig. 4a, Supplementary Fig. 48 and Supplementary Table 5). And the formation enthalpy of a single atom ($\Delta H$) for O-substituted $(NiCo)S_{0.89}$, $(NiCo)S$, $(NiCo)S_{1.33}$ and $(NiCo)S_2$ are calculated to be negative (Supplementary Table 6), indicating a thermodynamically stable structure after S-O substitution process. More importantly, the minimum $\Delta H$ of O-substituted $(NiCo)S_{1.33}$ further revealed that the energy required to replace the lattice sulfur S in $(NiCo)S_{1.33}$ with O atom is the lowest (Fig. 4b). These results suggest that the lattice sulfur in $(NiCo)S_{1.33}$ is more easily substituted by O atoms under the action of the applied voltage to form a new surface structure of $(NiCo)O_xS_{1.33-x}$.

Furthermore, the PDOS of O-substituted $(NiCo)S_{1.33}$ reveals an obvious electronic structure rearrangement due to the lattice sulphur-oxygen substitution (Supplementary Figs. 49–51). As a consequence, the d-band centre of metal atoms in $(NiCo)S_{1.33}$ has shown an upper shift while the other three sulphides exhibited various degrees of moving down (Fig. 4c, d, Supplementary Figs. 52–55 and Supplementary Tables 7, 8). And the d-band centre of Ni shifted more pronounced compared to that of Co, which can be attributed to the aggregated density of states of the dxz and dz$^2$ orbital for Ni 3d near the Fermi level (Supplementary Fig. 50g). According to previous work[40], Hammer and Nørskov demonstrated that the adsorption energy of small molecules on transition metals is correlated with the average energy of $\varepsilon_d$. The d-band centre has been usually used to describe the interaction strength between intermediates and catalysts' surface. On the basis of d-band theories, the higher d states in energy relative to the Fermi level, the less d-σ antibonding orbitals are occupied, indicating stronger adsorption bond[41]. Furthermore, the DFT calculation was utilised to investigate the adsorption energy of OH* on catalytically active Co and Ni sites. As shown in Supplementary Fig. 56, the adsorption energy of OH* on Co sites on (110) terminated surfaces of $(NiCo)S_{1.33}$ and O-substituted $(NiCo)S_{1.33}$ were calculated to be −2.02 and −2.11 eV, respectively. In comparison, the adsorption energy of OH* on Ni sites of both $(NiCo)S_{1.33}$ and O-substituted $(NiCo)S_{1.33}$ were lower than that on Co sites, which suggested that Co is the dominated active site. Besides, the adsorption energy of OH* on Ni sites was revealed to lift more obviously than that of Co, which implied the enhancement of OH* adsorption capacity of Ni. Thus, the upper shift of d-band centre in $(NiCo)S_{1.33}$ represents an enhancement of capacity for adsorbing the reaction intermediates during the OER process. And the significant improvement of Ni suggests that the Co-dominated active site of the intrinsic sulphide surface may convert into a Ni-Co double-active site due to the formation of $(NiCo)O_xS_{1.33-x}$ surface.

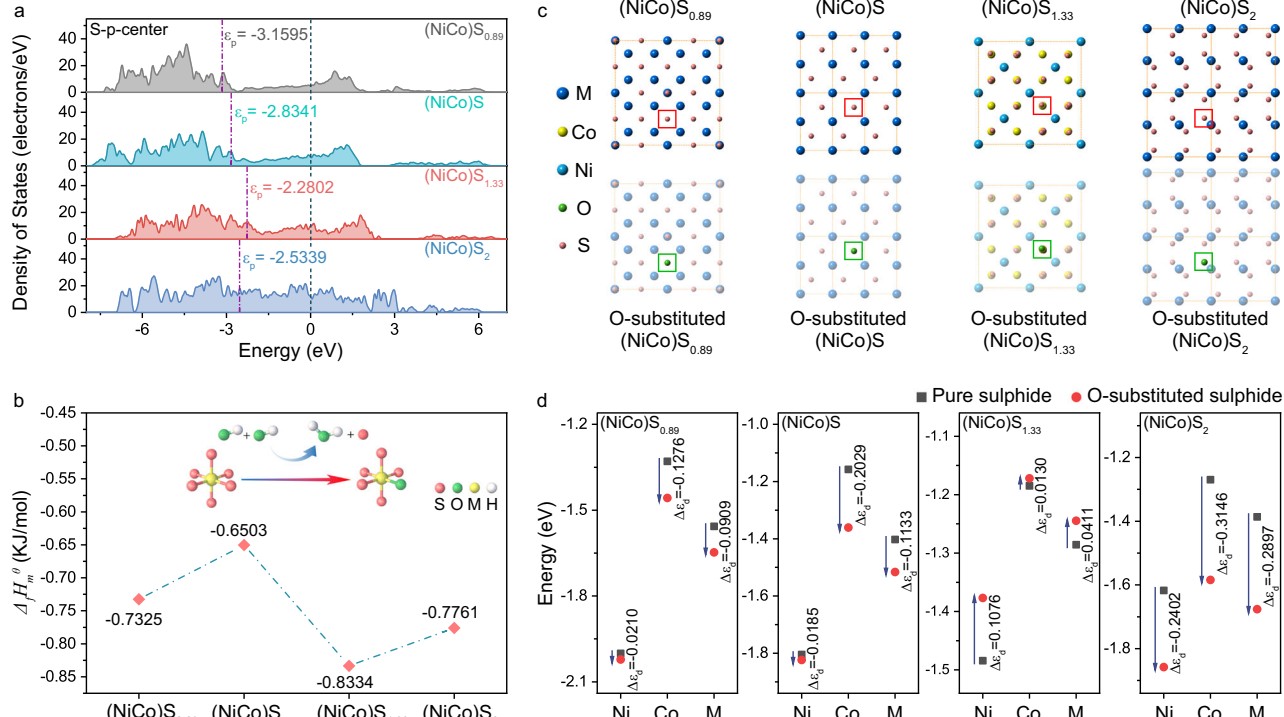

**Fig. 4 | Origin and influence of the lattice sulphur-oxygen substitution process.** **a** p-band centre of S in $(NiCo)S_{0.89}$, $(NiCo)S$, $(NiCo)S_{1.33}$ and $(NiCo)S_2$. **b** Formation enthalpy of replacing the lattice sulphur S with O atom in $(NiCo)S_{0.89}$, $(NiCo)S$, $(NiCo)S_{1.33}$ and $(NiCo)S_2$. **c** Schematic diagram of the pure sulphides and corresponding lattice O-substituted sulphides. S atom in pure sulphide is labelled with a red circle. O atom in O-substituted sulphide is labelled with a green circle. **d** Comparison of the metal d-band centre in the pure sulphides and corresponding lattice O-substituted sulphides.

## Influence of substitution process on OER mechanism

The S-O substitution process at the pre-catalytic stage of $(NiCo)S_{1.33}$ has shown the difference in the reconstruction process between these sulphides, which implies the different OER reaction mechanisms. Thus, we utilised in-situ Fourier transform infrared (FT-IR) spectroscopy measurements have revealed the influence of mechanism pathway on the adsorption of reaction intermediates[42–44] (Fig. 5a, b and Supplementary Fig. 57). The relative vibration intensity of adsorbed $O^*$ (M-O*) and $OH_{ads}$ (M-$OH_{ads}$) on the surface of $(NiCo)S_{1.33}$ is obviously lower than that of adsorbate evolving mechanism (AEM)-dominated $(NiCo)S_{0.89}$, $(NiCo)S$ and $(NiCo)S_2$, which is regulated by the lattice oxygen oxidation mechanism (LOM)-enhancing effect (Fig. 5c). Furthermore, in-situ differential electrochemical mass spectroscopy (DEMS) measurements[45] with the isotope $^{18}O$ was employed to distinguish the discrepancy of reaction mechanisms occurred on these sulphides during OER process (Supplementary Fig. 58). The ratio of LOM pathway is enhanced by ~240% on $(NiCo)S_{1.33}$ compared to $(NiCo)S_{0.89}$, $(NiCo)S$ and $(NiCo)S_2$ (Fig. 5d). This result indicated that the majority of the evolved $O_2$ was generated through the LOM on the surface of $(NiCo)S_{1.33}$, while the AEM plays a dominant role in the other three sulphides. At this stage, we can basically confirm that the S-O substitution process of $(NiCo)S_{1.33}$ acts as a pre-treatment process which induces a $(NiCo)O_xS_{1.33-x}$ surface to enhance the ratio of the LOM pathway and further promotes the surface reconstructing into metal-(oxy)hydroxides to drive OER (Fig. 5e). In contrast, the direct reconstruction of other sulphides leads to the AEM dominated mechanism.

## Comparison of the $(NiCo)O_xS_{1.33-x}$ surface and pure oxide surface

In order to further investigate the advantages of $(NiCo)O_xS_{1.33-x}$ surface during the OER process, a pure oxide of $(NiCo)O_{1.33}$ with the same morphology and size was selected as the contrastive pure oxide surface. As shown in Fig. 6a, the $(NiCo)O_xS_{1.33-x}$ surface exhibits higher

catalytic activity than the $(NiCo)O_{1.33}$ surface. In addition, the catalytic activity of the pure oxide surface exhibited a decrease after five CV cycles, while the $(NiCo)O_xS_{1.33-x}$ surface showing an increase trend (Fig. 6b), implying an enhancement effect of the S-O substitution process on the catalytic performance. Moreover, the series of atomic in-situ HAADF images (Fig. 6c) under different applied potentials have shown that the surface reconstruction of $(NiCo)O_{1.33}$ is lagging and weaker compared to $(NiCo)S_{1.33}$. Moreover, the in-situ EELS of O K-edge (Fig. 6d) revealed that the voltage required to drive the lattice oxygen in the pure oxide surface to participate in the reaction is much higher than that in the $(NiCo)O_xS_{1.33-x}$ surface. This result suggests that the lattice oxygen obtained through the S-O substitution process exhibits a higher reaction activity than that of the intrinsic lattice oxygen in the corresponding oxide.

According to the in-situ EELS $L_3$-edge spectrum of $(NiCo)S_{1.33}$ (Fig. 6e and Supplementary Fig. 59) and $(NiCo)O_{1.33}$ (Fig. 6f and Supplementary Fig. 60), the threshold voltage of the metal-valence state elevation occurred in $(NiCo)S_{1.33}$ is significantly advanced due to the addition of the S-O substitution process. Moreover, the Ni $L_3$ edge of $(NiCo)S_{1.33}$ exhibited a more obvious shift to the high-energy direction compared to Co under the applied potential of 0.9 V. However, the valance state of Ni in $(NiCo)O_{1.33}$ was revealed to slightly elevate until the applied voltage increased to 1.1 V, while the obvious elevation of Co occurred at 1.0 V. This result has revealed that the applied potential for the hydroxide ion intercalation induced surface reconstruction on the $(NiCo)O_xS_{1.33-x}$ surface is much lower than that of the pure oxide surface, which explained the intrinsic reason that the catalytic activity of $(NiCo)S_{1.33}$ is significantly higher than that of $(NiCo)O_{1.33}$. Furthermore, the characteristic peaks of M-OH[46–49] in $(NiCo)O_{1.33}$, which were extracted from the operando Raman spectra (Supplementary Fig. 61), should mainly come from the contribution of Co-OH due to the great gap between the Co-OOH[50,51] and Ni-OOH[52,53], indicating that Co is suggested to be the dominant metal active site in the pure oxide

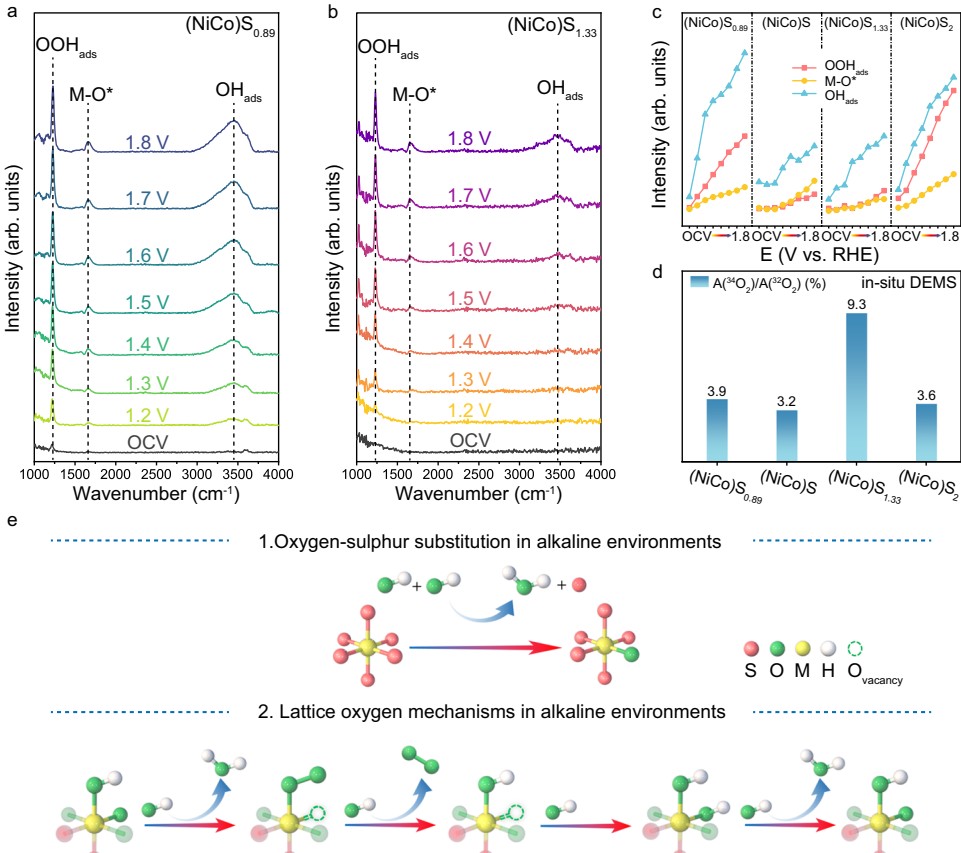

**Fig. 5 | Influence of substitution process on OER mechanism. a, b** In-situ FT-IR spectra of $(NiCo)S_{0.89}$ and $(NiCo)S_{1.33}$ were recorded during the multi-potential steps. **c** Signal strength variation diagram of characteristic FT-IR peaks of $OOH_{ads}$, $M-O^*$ and $OH_{ads}$. **d** The ratio of the LOM pathway extracted from DEMS signals of $^{34}O_2$ and $^{32}O_2$ of $(NiCo)S_{0.89}$ and $(NiCo)S_{1.33}$ from the reaction products cycled in $H_2^{16}O$ aqueous KOH electrolyte. **e** Schematic diagram of the lattice oxygen mechanism with lattice sulfur-oxygen substitution process.

surface for capturing OH* and promote the occurrence of OER reaction, while Ni plays an auxiliary role. In comparison, the applied voltage leading to the occurrence of Co-OOH and Ni-OOH characteristic peaks in $(NiCo)S_{1.33}$ is obviously lower than that in $(NiCo)O_{1.33}$. And the gap of relative strength between Co-OOH and Ni-OOH in $(NiCo)S_{1.33}$ is smaller than that of $(NiCo)O_{1.33}$. This promotion indicates a double-metal active site in the $(NiCo)O_xS_{1.33-x}$ surface, which could be attributed to the pronounced upper shift of Ni d-band centre-induced enhancement of adsorption capacity for OH* (Supplementary Fig. 56) due to the substitution process at the pre-catalytic stage. Meanwhile, the characteristic peaks of M-OH, Co-OOH and Ni-OOH of $(NiCo)S_{1.33}$ showcased a more sensitive response to the applied voltage than that of $(NiCo)O_{1.33}$ (Fig. 6g, h), indicating a lower threshold voltage of the ion intercalation. These results revealed that the $(NiCo)O_xS_{1.33-x}$ surface can enhance the capacity of Ni for capturing OH*[54] and further accelerate the surface reconstruction to exhibit a higher promotion effect on the OER process compared to the pure oxide surface.

In conclusion, we realised a direct imaging of the dynamic phase evaluation pathway at the pre-catalytic stage of NiCo-based sulphides before they reconstruct into active species as the metal-oxy/hydroxide under alkaline OER operation. The lattice sulphur atoms on the surface of $(NiCo)S_{1.33}$ particles are partially substituted by oxygen in electrolyte with a maintained crystal structure under the pre-catalytic potential of 0.9 V, which further induces the formation of a lattice oxygen-sulphur coexisting surface as $(NiCo)O_xS_{1.33-x}$. The palingenetic $(NiCo)O_xS_{1.33-x}$ surface can significantly lift the LOM ratio of the sulphide matrix to a comparable level as its corresponding oxide and contribute to a higher catalytic activity, which reduces energy barrier

for superficially reconstructing into the NiCo-oxy/hydroxide as the active electrocatalyst to further boost OER process. Our work has shown a comprehensive pathway for elaborating the dynamic phase evolution of bimetallic sulphides at the pre-catalytic stage and compactly linking the internal mechanism of phase transition with the atomic occupation forms of metal atoms, which completes the missing puzzle of the catalytic mechanism of sulphides.

## Methods
### Synthesis of cobalt-nickel bimetallic sulphides
The details of synthesis method of $NiCo_2O_4$ precursor are expounded in the Experimental section of Supplementary Information. Then, the $NiCo_2O_4$ precursor were used to synthesise the cobalt-nickel bimetallic sulphides by adjusting the evaporation rate of sulfur powder based on temperature control, 300 mg sulfur powder and the above $NiCo_2O_4$ NWs were placed in two crucibles, which are 20 cm apart in the tube. Then the tube was heated to 300, 500, 600 and 800 °C with a rate of 10 °C min⁻¹ for 2 h under the flowing Ar atmosphere. Finally, the system was cooled under a flowing Ar atmosphere and the $(NiCo)S_2$, $(NiCo)S_{1.33}$, $(NiCo)S$ and $(NiCo)S_{0.89}$ nanocrystals could be obtained, respectively.

### Electrochemical measurements
All of the electrochemical measurements were performed by using a CHI760E Electrochemical Workstation (CHI Instruments, Shanghai Chenhua Instrument Corp., China) in $O_2$-saturated 1.0 M KOH solution with a typical three-electrode setup. A Pt-foil and Hg/HgO with 1.0 M KOH filling solution were used as the counter and reference electrodes,

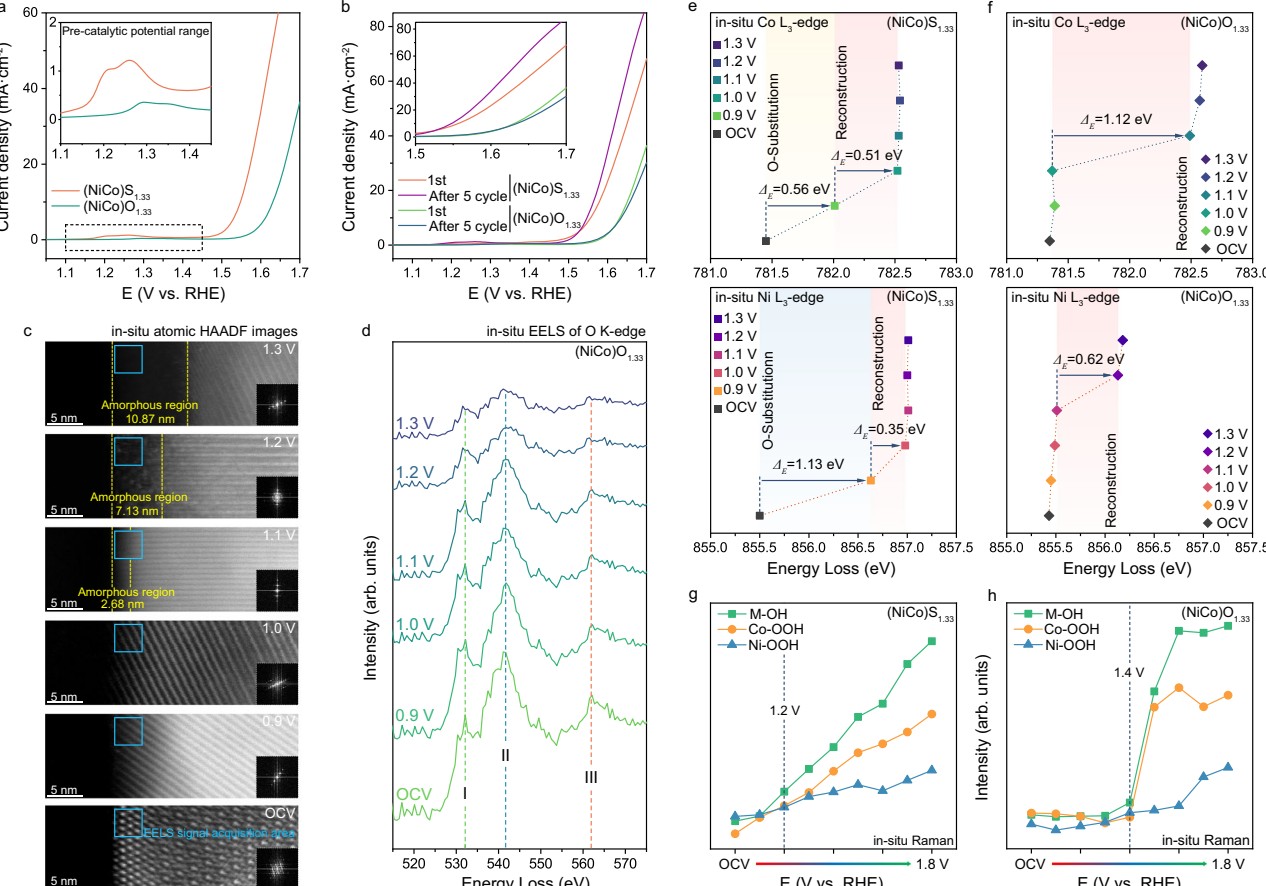

**Fig. 6 | Comparison of the (NiCo)O$_x$S$_{1.33-x}$ surface and the pure (NiCo)O$_{1.33}$ surface. a** iR-corrected LSV at a scan rate of 5 mV s$^{-1}$ of (NiCo)S$_{1.33}$ and (NiCo)O$_{1.33}$. **b** iR-corrected LSV of (NiCo)S$_{1.33}$ and (NiCo)O$_{1.33}$ before and after five CV cycles at a scan rate of 5 mV s$^{-1}$. **c** In-situ atomic HAADF images of (NiCo)S$_{1.33}$ after constantly applying different potentials for twenty seconds. **d** In-situ EELS spectra of O K-edge under different applied potential. The I, II and III peaks can be assigned to the hybridisation of O 2$p$ with Co 3$d$, Ni 3$d$ and Co 4$sp$ orbitals, respectively. **e, f** Peak position of Co and Ni L$_3$-edge collected from the EELS spectra of (NiCo)S$_{1.33}$ and (NiCo)O$_{1.33}$ L-edge. **g, h** Signal strength variation diagram of characteristic Raman peaks of M-OH, Co-OOH and Ni-OOH of (NiCo)S$_{1.33}$ and (NiCo)O$_{1.33}$.

respectively. The as-measured potentials (versus Hg/HgO) were calibrated with respect to the RHE. All electrochemical experiments were conducted under the temperature of 20 ± 0.2 °C. A glassy carbon electrode with a diameter of 3 mm covered by a thin catalyst film was used as the working electrode. The catalysts were prepared by dispersing 3 mg of catalyst@C in 1450 μL of N, N-dimethylformamide (DMF) with 50 μL of 5 wt% Nafion solution. Then 7.1 μL of the catalyst ink was loaded onto the surface glassy carbon electrode (mass loading: 0.02 mg cm$^{-2}$) and air-dried at room temperature. For cyclic voltammetry (CV) measurements, the scan rate was set to 10 mV s$^{-1}$. To reduce the influence of capacitive current and gas bubbles, the CV cycling process was performed with a chronoamperometry between 1.0 and 1.7 V$_{RHE}$ at a pulse width of 10 s. Every 100 cycles, a cyclic voltammetry between 1.0 and 1.7 V$_{RHE}$ at 10 mV s$^{-1}$ has been recorded. In order to compensate for the effect of solution resistance, the potentials were corrected by using the following equation: E$_{iR\ corrected}$ = E - iR, in which R is the uncompensated ohmic resistance of the solution. All voltages in this work are referenced against the RHE unless stated otherwise.

### In-situ Infrared and Raman spectroscopy
Operando FT-IR measurements were conducted on a BRUKER-Fourier Transform Infrared Spectrometer-TENSOR27. A stability potential was applied to the catalyst electrode for 10 min before carrying out all infrared spectral. Before each OER measurement of the system, the background spectrum of the catalyst electrode was obtained under the open circuit voltage, and the measurement potential of OER was in

the range of 1.2–1.8 V with an interval of 0.1 V. Operando Raman spectroscopy measurements were conducted on a LabRAM HR Evolution spectrophotometer with 532 nm wavenumber of the excitation light source. Under each applied voltage, the system was stabilised for 5 min before the measurements.

### In-situ transmission electron microscope
A Protochips liquid cell holder (Poseidon Select 550) was used for the in-situ EC-TEM and EELS analysis. The working electrode, counter electrode and reference electrode on the electrochemical chip used for in-situ experiment are all made of Pt. The electrochemistry chip was washed in HPLC-grade acetone, methanol, and ethanol for 2 min to remove its protective coating before loading the catalysts. Then, the membrane side was plasma cleaned for 2–5 min with an Ar/O$_2$ mixture to render it hydrophilic. Once the plasma is cleaned, the surface will remain typically remain hydrophilic for several hours. After that, solvent-dispersed samples of (NiCo)S$_{1.33}$ was placed directly onto the silicon nitride observation window of the bottom (spacer) E-chip using a pipette. Particularly, in order to obtain a clear field of vision and prevent a large number of samples from piling up and damaging the silicon nitride film of the chip during the in-situ experiment, we only dropped a small amount of pure sulphide nanoparticles on the chip. Before the closure of the in-situ cell, the chip was dried under air by baking for 1 min at a temperature of 50 °C to ensure that the catalysts were tightly adsorbed on the working electrode. The electrolytes (0.1 M KOH) were flushed into the in-situ cell after its introduction into

the microscope at a constant 1 µL min⁻¹ flow. Electrochemical experiments were performed with a Gamry 600+ potentiostat in an aberration-corrected scanning transmission electron microscope (FEI Titan Cubed Themis G2 300, FEI, USA) operated at 300 kV, operated in continuous capture mode to yield image sequences of the dynamics during electrocatalysis. The scan rate to 100 mv/S during the in-situ CV test. The movie was recorded at 10 frames per second in STEM mode.

## Data availability

The data that support the plots are available within this paper and its Supplementary Information. All other relevant data that support the findings of this study are available from the corresponding authors on reasonable request. Source data of Figures in the main text and Supplementary Information are provided as supplementary files. Source data are provided with this paper.

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

## Acknowledgements

We acknowledge support from the National Natural Science Foundation of China (No. 21922105 and 21931001 received by P.X. and 22205088 received by Y.H.), the National Key R&D Programme of China (2021YFA1501101 received by P.X.), the Special Fund Project of Guiding Scientific and Technological Innovation Development of Gansu Province (2019ZX-04 received by P.X.) and the 111 Project (B20027 received by P.X.). We also acknowledge support by the Fundamental Research Funds for the Central Universities (lzujbky-2021-pd04, lzujbky-2021-it12 and lzujbky-2021-37 received by P.X.).

## Author contributions

Y. Hu, Y.Z., P.X. and C.-H.Y. conceived the original concept and initiated the project. Y. Hu, J.J. and J.Y. prepared the materials and performed electrochemical and XRD measurements and analysed electrochemical, Raman, FT-IR and DEMS data. L.A. and M.L. analysed XAFS data. W.S. carried out in-situ Raman and FT-IR measurements. Y. Hou carried out the DEMS measurements. Y.W. worked on the DFT calculations and analysis. Y. Hu, Y.P. and L.Z. designed and carried out the structural characterisation and in-situ electron microscopy investigations. Y. Hu analysed the STEM and EELS data. Y. Hu wrote the manuscript with input from all authors. Y.Z., P.X. and C.-H.Y. revised the manuscript.

## Competing interests

The authors declare no competing interests.
