## [Peer Review File · Nature Communications]

REVIEWER COMMENTS

Reviewer #1 (Remarks to the Author):

The authors prepared a series of Co-Ni bimetallic sulphides in this work and investigated the phase transition mechanism during OER. In situ TEM, FTIR and Raman were employed to elucidate the oxygen-substitution phenomenon. Besides, DFT calculation was conducted as a theoretical explanation for the superior performance of (NiCo)S_{1.33}. This work gives a systematic investigation of the phase evolution of Co-Ni sulphides and is suggested to be published in Nature Communication after addressing the following issues:

1. The CV curves in Supplementary Fig. 21 (Fig. S21) are weird. Why are the edges so sharp but not smoothly turned?
2. In Fig. S24, the oxidation peaks are given through both in-situ and ex-situ techniques. The oxidation peak of Ni is higher than Co under ex-situ CVs, while why is it reversed under in-situ conditions? Besides, the authors used Pt as a reference electrode; what's the real potential vs RHE? Can it be converted to RHE potential?
3. According to the pre-catalytic range, the authors applied 0.9 V voltage (line 149) to ensure full oxidation and observed the growth of a new shell. This is a normal phenomenon as the particle is undoubtedly oxidized. However, what if a lower voltage is applied, e.g., V in the pre-catalytic range or even lower? Since in-situ TEM is able to capture the atomic change, is it possible that the surface is already oxidized even at low voltage before the pre-catalytic stage?
4. In line 281, the authors claimed that "the Ni-OOH signal in (NiCo)S_{1.33} has exhibited a significant enhancement....." which may not be appropriate. As indicated in Fig. S45, the relative peak intensity between CoOOH and NiOOH I_{Co}/I_{Ni} in both (NiCo)S_{1.33} and (NiCo)O_{1.33} are ~2. It is insufficient to prove the promoted capacity of Ni.
5. Some other errors need revision, e.g., in Figure 4c, the schematic diagram of the pure sulphides seems to be over-cropped without complete atoms (on the middle line); the legend for Fig. 4d is missing; the notes under Fig. S23 is placed wrong; in line 243, LOM has to be written out in full the first time used.

Reviewer #2 (Remarks to the Author):

This manuscript documents a study on revealing the lattice sulfur-oxygen substitution of metal sulfide electrocatalysts at pre-catalytic stage during oxygen evolution reaction. Overall the research was adequately designed with suitable methodology and the results are interesting. I would recommend the following major revisions before the manuscript being considered for publication:

1) In Fig. 2 and also in the video clip, as the in-situ TEM experiment continues up to more than 30s, there are obvious total mass increase, particle coalition and particle agglomeration at the vicinity of the working electrode. Please specify the following:

a) Can you exclude Ostwald ripening process in this electrochemical environment? If so, how?

b) Instead of using a field of view with many (small) overlapped flakes, can you please choose one with less and clearly defined individual particles and present more convincing results?

2) In Fig. 3, are the results including the HRTEM images in Fig. 3a truly collected in an in-situ environment with a flow of fresh KOH liquid and varied potentials constantly applied? If so, what's the estimated thickness of KOH in the liquid cell? How does KOH affect corresponding HRTEM, EELS and EDX results as shown in this figure?

3) Again in Fig. 3, Can you have a quantitative estimation of the relative atomic concentrations of O and S at 0.9V for 20S and does it match a lattice reconstruction consistent with the HRTEM result?

4) How about a possible in-situ EC-TEM experiment and results with reversed potentials following the initial amorphous shell formation, or the structural evolution of relevant phases during I-V loop?

5) There are sporadic obvious typos and grammatical errors to be corrected. For example:

a) line 31 "direction observation" => "direct observation"?

b) line 71 "...that driving..." => "...that drives..." or "...driving..."

Please proofread thoroughly.

Reviewer #3 (Remarks to the Author):

In this manuscript the authors have investigated the surface structure evolution of Ni-Co-S of varied stoichiometry under conditions of OER. Recently transition metal chalcogenides have attracted significant attraction as electrocatalysts for various electrochemical processes including OER, ORR, and HER. Among these the performance of transition metal chalcogenides, especially, selenides and tellurides have shown tremendous promise for OER exhibiting some of the lowest overpotentials reported till date. In spite of their exceptional activity, there is still confusion regarding the actual catalytic species for these compositions. While traditionally it has been proposed that the chalcogenides are fully converted to (oxy)hydroxide surfaces under conditions of OER, there has been recent studies also which confirms that the chalcogenides actually form a mixed anionic (hydroxy)chalcogenide (selenide or telluride) surface, where the metal is coordinated to both Se/Te and OH.[1,2] This manuscript constitutes an important step in that direction of gathering evidence for structural evolution of the surface under conditions of OER. However, the authors have only considered the least active (OER) and least stable chalcogenide, i.e. sulfides for their study. The inherent instability of the sulfides towards hydrolysis and even under ambient conditions dilutes the importance of these results, such that they cannot be extended to other highly active chalcogenides. What could have potentially been transformative for the field has become very system-specific without any far-reaching implications. This could have been important observations, however the narrow scope of these results along with other technical and scientific flaws makes it not ready for publication yet (in my opinion).

Major concerns:

- Scrambling of the catalytic site: Ni vs Co

The authors have proposed that Co is the actual catalyst site in the most active composition (NiCo₂Se₄) although Ni assists to boost performance. However, from the DFT calculation it was shown that the Ni d-band center was higher and closer to the Fermi surface in the band structure diagram. This would imply -OH binding to the Ni-site (d-orbitals) as opposed to Co, making Ni as the catalytic site. This is also supported by the observation that Ni- pre oxidation peak precedes the OER onset, which suggests that Ni pre-oxidation constitutes catalyst activation. If Co was the actual catalytic site, OER onset will not follow the Ni-site activation step.

In fact in the mixed metal chalcogenides, understanding of scrambling of the catalyst sites is a major challenge. The authors may utilize DFT studies to actually investigate the site activation step between Co and Ni and correlate that with the observed (bulk) overpotential to offer better insight into the actual catalyst site.

- Except for NiCo₂Se₄ the other 3 compositions reported in this manuscript are mixtures of binary sulfides of Ni-S and Co-S. It is not advisable to compare activity of a stoichiometric mixed metal selenides with mixture of binaries, especially when both the binaries (Ni-S and Co-S) are active for OER to varied extent. Mixture of binaries will expectedly lead to performance which is either

representative of the average of the individual components (Ni-S and Co-S phases) or the dominant one.

- The Co pre-oxidation peak is typically observed at ~ 1.1 v vs RHE. However, this peak was significantly in the LSV plots of even the bulk samples. Does that signify that Co-site is not getting oxidized under these conditions. If so, then Co is not the actual catalytic site and the premise of discussion in the manuscript is wrong.

- Although the premise of in situ electrochemical studies on microscopic sample space is intriguing, I have a major concern regarding the effect of the accelerated e-beam (from TEM) on the observed electrochemical behavior. The e-beam will lead to surface impingement of large density of electrons that will be dissipated throughout the sample at various rates depending on the intergrain conductivity. NiCo₂Se₄ with higher conductivity will dissipate the electrons faster leading to larger current density. However, that current density is not necessarily from OER but rather from facile electron transfer across the catalyst surface. The binary sulfides on the other hand, will lead to sluggish growth of current density since the large density of impinged electrons are trapped on the surface. The authors should perform more control experiments including the one where the current density is recorded under full exposure but zero bias.

- The other concern regarding accelerated e-beam exposure is local surface heating where it has been observed previously that local temperature of the beam spot can go as high as several hundred of degrees. Sulfides are known to degrade by loss of S atoms at elevated temperatures. In fact, prolonged TEM imaging of sulfides under normal conditions (high vacuum and no reactive atmosphere exposure) also lead to such decomposition and loss of S. Hence, it might be wrong to assume that the S loss observed in these TEM imaging is due to progression of OER. The authors should also study S leaching (if any) from the bulk samples through analytical techniques, quantify that and then correlate with the electron microscopy studies.

Pinxian Xi

State Key Laboratory of Applied Organic Chemistry
College of Chemistry and Chemical Engineering
Lanzhou University
Lanzhou 730000, China.
E-mail: xipx@lzu.edu.cn

Jan. 30th, 2023

Manuscript ID NCOMMS-22-39322-T

Dear Reviewers,

Thank you for the important comments on our paper “*Revealing the Lattice Sulphur-Oxygen Substitution Pathway of Metal Sulphide Electrocatalysts at Pre-catalytic Stage during Oxygen Evolution Reaction*” (Manuscript ID NCOMMS-22-39322-T). We appreciate your efforts and time in reviewing our manuscript. Your comments are positive and constructive, which are very helpful for us to further improve our work. Based on these comments and suggestions, we have revised the paper, and all changes made in the revised manuscript are highlighted in yellow. Our answers to Reviewers’ questions are given below:

To Reviewer 1:

General comments: The authors prepared a series of Co-Ni bimetallic sulphides in this work and investigated the phase transition mechanism during OER. In situ TEM, FTIR and Raman were employed to elucidate the oxygen-substitution phenomenon. Besides, DFT calculation was conducted as a theoretical explanation for the superior performance of (NiCo)S_{1.33}. This work gives a systematic investigation of the phase evolution of Co-Ni sulphides and is suggested to be published in Nature Communication after addressing the following issues:

General response:

We sincerely appreciate your valuable comments. We have synthesized new reference materials and performed additional characterizations to address the Reviewer’s concerns and support our conclusion.

Briefly, in response to the comments we have:

- Clarified the cause of the sharp edge in the CV curves and technical details of the *in-situ* experiment.
- Captured the changes of crystal structure, element distribution and metal-oxygen bonds on the surface of (NiCo)S_{1.33} under lower potential.
- Verified the promoted capacity of Ni for capturing OH* by DFT calculation.
- Corrected the errors in writing and graphics.

Specific responses to the comments and corresponding modifications are provided below.

Q1. *The CV curves in Supplementary Fig. 21 (Fig. S21) are weird. Why are the edges so sharp but not smoothly turned?*

A1. Thank you for your carefully reading and pointing this out. The CV curves in Fig. S21 are used to measure the electric double layer capacitance for further calculating the electrochemical active area of different sulphides (as shown in Fig. S20 c). In the actual testing process, we first measured the open circuit potential of different sulphides, and then set the potential range as $OCP \pm 0.05$ V with the scan rate of 2, 4, 6, 8 and 10 mV/s, respectively. For example, the range of applied potential was set as 0.85 to 0.95 V (V vs. RHE) for (NiCo)S_{0.89}. This potential range is in the non-Faraday current range, only the double layer charging and discharging process occurs, without any electrochemical reaction. And the CV curve will exhibit a rectangular-like shape. Therefore, the edge of CV curves in Fig. S21 are sharp, which is different from the smooth conversion of conventional CV curves (Fig. S22) with complete electrochemical reaction. Please find that we have added this part of exposition in the revised supplementary information at page 25.

Q2. *In Fig. S24, the oxidation peaks are given through both in-situ and ex-situ techniques. The oxidation peak of Ni is higher than Co under ex-situ CVs, while why is it reversed under in-situ conditions? Besides, the authors used Pt as a reference electrode; what's the real potential vs RHE? Can it be converted to RHE potential?*

A2. Thank you for your carefully reading and pointing this out.

Firstly, the working electrode, counter electrode and reference electrode on the electrochemical chip used for *in-situ* experiment are all made of Pt (Fig. R1). For *ex-situ* electrochemical measurement, the glassy carbon electrode, Pt-foil and Hg/HgO with 1.0 M KOH filling solution are used as the working, counter and reference electrodes, respectively.

Electrochemical chip

Working electrode

Fig. R1. Schematic diagram of electrochemical chip.

Secondly, in order to obtain a smooth and ideal curve, we have set the scan rate to 100 mV/S during the *in-situ* CV test. The parameter used in *ex-situ* electrochemical test is 10 mV/S (see **Electrical Measurements** in **Method** for details).

Thirdly, in order to obtain a clear field of vision and prevent a large number of samples from piling up and damaging the silicon nitride film of the chip during the *in-situ* experiment, we only dropped a small amount of pure sulphide nanoparticles on the chip, which results in a very few samples being loaded on the working electrode. However, the catalysts for *ex-situ* test were prepared by dispersing 3 mg of catalyst@C in 1450 μ L of N, N-Dimethylformamide (DMF) with 50 μ L of 5 wt% Nafion solution.

Therefore, we suggest that the three differences between the *in-situ* and *ex-situ* CV tests leads to the reversal of the relative strength of Ni and Co oxidation peaks. However, we believe that the intrinsic structure, elemental distribution and valence transition of catalysts should be consistent whether under *in-situ* or *ex-situ* conditions. The phenomena of surface oxygen sulfur exchange and surface reconstruction observed by *in-situ* experiments can reflect the actual electrochemical behavior of sulphides in the OER process. Besides, the main purpose of *in-situ* CV curve is to determine the voltage range of the pre-catalytic stage and provide reference for setting applied potential for subsequent *in-situ* IT experiments.

And regarding your question about converting the voltage value of the *in-situ* test to the real potential vs RHE, we have carefully consulted and discussed with the application engineers of Protochips. Unfortunately, since the working, counter and reference electrode on the chip are made of Pt, there is no conversion formula or corresponding theory that can accurately convert the voltage in the *in-situ* experiments to RHE potential. Sorry for not directly answering your question about voltage conversion, but we suggest that the 0.9 V under *in-situ* condition corresponds to 1.4 V under *ex-situ* condition by

comparing the applied potentials at the end of the pre-catalytic stage under *in-situ* and *ex-situ* conditions. And please find that we have rewritten the “Platinum was used as the reference electrode, RE” in line 147 as “Pt pseudo-Reference”, which should be more rigorous, and added the description of technical details and corresponding discussions at the *in-situ* **Transmission Electron Microscope** part of **Methods** in the revised manuscript and page 29 in the revised supplementary information, respectively.

Q3. *According to the pre-catalytic range, the authors applied 0.9 V voltage (line 149) to ensure full oxidation and observed the growth of a new shell. This is a normal phenomenon as the particle is undoubtedly oxidized. However, what if a lower voltage is applied, e.g., V in the pre-catalytic range or even lower? Since in-situ TEM is able to capture the atomic change, is it possible that the surface is already oxidized even at low voltage before the pre-catalytic stage?*

A3. Thank you for your thoughtful comments and important suggestion.

At the pre-catalytic stage, under the premise that the crystal structure of $(\text{NiCo})\text{S}_{1.33}$ surface remains spinel structure, the sulphur atoms are partially substituted by oxygen atoms to form the $(\text{NiCo})\text{O}_x\text{S}_{1.33-x}$ structure before the catalytic surface fully reconstructing into active metal-oxy/hydroxide. This is different from the oxidation behavior of the other three sulphides, which are accompanied by a serious loss of S and the direct transformation of the original crystal structure to metal-oxy/hydroxide at the pre-catalytic stage. According to your suggestion, we have captured the changes of the crystal structure, element distribution and metal-oxygen bonds on the surface of $(\text{NiCo})\text{S}_{1.33}$ under lower potentials.

According to the atomic HAADF images (Fig. R2 a), element mapping (Fig. R2 b) and corresponding element distribution curves (Fig. R2 c), it could be found that the $(\text{NiCo})\text{S}_{1.33}$ surface still maintained good crystallinity and no enrichment of oxygen element was detected under the lower potential conditions after 20 seconds. Combined with O-EELS, it can be confirmed that there is no generation of metal-oxygen bond on the $(\text{NiCo})\text{S}_{1.33}$ surface under the lower potentials. Therefore, we suggest that the lattice sulfur-oxygen substitution process of $(\text{NiCo})\text{S}_{1.33}$ surface is occurred under 0.9 V at the pre-catalytic stage. In the other words, the $(\text{NiCo})\text{S}_{1.33}$ surface is not oxidized at low voltage.

Fig. R2. Structural, elemental evolution and O K-edge of (NiCo)S_{1.33} under different applied potentials.

After carefully considering your suggestion, we added an *in-situ* IT experiment under the applied voltage of 0.9 V for 10 seconds to further reveal the changes that occur on the (NiCo)S_{1.33} surface at 0.9 V. According to Fig. R3 a, b and c, it could be found that the (NiCo)S_{1.33} surface had a certain degree of oxygen enrichment with the constant crystal structure, and the width of the oxygen enrichment area is 3.68 nm (Fig. R3 c and d). Combined with O-EELS (Fig. R3 e) and the change of sulphur-oxygen content under different voltages (Fig. R4 and Table R1), it can be confirmed that M-O bonds were formed on the surface. These results have revealed that, the formation of sulphur-oxygen coexistence ((NiCo)O_xS_{1.33-x}) surface started after the applied voltage rising up to 0.9 V due to the sulfur-oxygen substitution process. And compared to the sulphur-oxygen coexistence surface with the thickness of 8 nm (Fig. 3c and d), which was generated at 0.9 V within 21 seconds, it could be found that the thickness of (NiCo)O_xS_{1.33-x} shell thickens with the increase of sulfur-oxygen substitution time.

Fig. R3. Structural, elemental evolution and O K-edge of $(\text{NiCo})\text{S}_{1.33}$ under the applied potential of 0.9 V after 10 s.

Fig. R4. Atomic fraction of S and O in $(\text{NiCo})\text{S}_{1.33}$ under the different applied voltage.

Table R1. The atomic fraction of elements in (NiCo)S_{1.33} under different applied voltage. The atomic fraction of Co, Ni, S and O were extracted from elemental mapping data.

(NiCo)S _{1.33}	Co	Ni	S	O
0.50 V	28.09	12.16	55.23	4.52
0.60 V	27.74	13.01	54.15	5.10
0.70 V	27.55	12.57	54.93	4.95
0.80 V	27.42	12.85	55.01	4.72
0.90 V (10 s)	33.86	13.13	43.28	16.08

The phenomena observed under the applied voltage of 0.9 V after 10 seconds is consistent with that of 0.9 V after 20 seconds, except for the difference in the thickness of the sulphur-oxygen coexistence surface. Thus, it could be determined that the sulfur-oxygen substitution process will start until the voltage is raised up to 0.9 V at the pre-catalytic stage. And none of the surface oxidation will occur when the applied voltage is lower than 0.9 V. Please find that these parts have been add as **Supplementary Fig. 45, 46, 47** and **Supplementary Table 4** in the revised supplementary information. And their corresponding discussions have been added at page 8 in the revised manuscript.

Q4. In line 281, the authors claimed that “the Ni-OOH signal in (NiCo)S_{1.33} has exhibited a significant enhancement.....” which may not be appropriate. As indicated in Fig. S45, the relative peak intensity between CoOOH and NiOOH I_{Co}/I_{Ni} in both (NiCo)S_{1.33} and (NiCo)O_{1.33} are ~2. It is insufficient to prove the promoted capacity of Ni.

A4. Thank you for your carefully reading and pointing this out.

The signal strength variation diagram of characteristic Raman peaks of M-OH, Co-OOH and Ni-OOH in Fig. 6g and h were extracted from the operando Raman spectra (Fig. S45), which was used to reflect the variation of relative intensity of different characteristic peaks with applied voltage. According to Fig. 6g, h and Fig. S45, the applied voltage leading to the occurrence of Co-OOH and Ni-OOH characteristic peaks in (NiCo)S_{1.33} is obviously lower than that in (NiCo)O_{1.33}. And the gap of relative strength between Co-OOH and Ni-OOH in (NiCo)S_{1.33} is smaller than that of (NiCo)O_{1.33}. Furthermore, the DFT results have revealed that the upper shift of Ni d-band center in (NiCo)S_{1.33} (Fig. 4d, Fig. S37 and Fig. S38) due to the sulfur-oxygen substitution process, which represents an enhancement of capacity

for adsorbing the reaction intermediates during OER process. And the Ni L₃ edge of (NiCo)S_{1.33} exhibited a more obvious shift to the high-energy direction and a lower threshold potential for shifting compared to that of (NiCo)O_{1.33}.

Follow your suggestion, we have utilized the DFT studies to calculate the adsorption energy of OH* on catalytically active Co and Ni sites respectively. The higher OH* adsorption energy indicates more preferential attachment on the active site. As shown in Fig. R5, the adsorption energy of OH* on Ni sites on (110) terminated surfaces of (NiCo)S_{1.33} and O-substituted (NiCo)S_{1.33} were calculated to be -1.68 and -1.92, respectively. In comparison, the adsorption energy of OH* on Co sites of both (NiCo)S_{1.33} and O-substituted (NiCo)S_{1.33} were higher than that on Ni sites. Besides, the adsorption energy of OH* on Ni sites was revealed to lift more obvious than that of Co, which implied the enhancement of OH* adsorption capacity of Ni.

Fig. R5. Comparison of OH* adsorption energy on catalytically active Co and Ni sites on (110) terminated surfaces of (NiCo)S_{1.33} and O-substituted (NiCo)S_{1.33}.

These results are combined to reach the conclusion that “the Ni-OOH signal in (NiCo)S_{1.33} has exhibited a significant enhancement, which suggested that the capacity of Ni for capturing OH* is promoted”. Your suggestion makes us find that the arguments for this conclusion are not comprehensive and accurate. Please find that these parts have been add as **Supplementary Fig. 56** in the revised supplementary information. And their corresponding discussions have been added at page 11 in the revised manuscript and page 61 in the revised supplementary information.

Q5. *Some other errors need revision, e.g., in Figure 4c, the schematic diagram of the pure sulphides seems to be over-cropped without complete atoms (on the middle line); the legend for Fig. 4d is missing; the notes under Fig. S23 is placed wrong; in line 243, LOM has to be written out in full the first time used.*

A5. Thank you for your carefully reading and pointing these errors out. Please find that these errors have been corrected in the revised manuscript.

To Reviewer 2:

General comments: *This manuscript documents a study on revealing the lattice sulfur-oxygen substitution of metal sulfide electrocatalysts at pre-catalytic stage during oxygen evolution reaction. Overall the research was adequately designed with suitable methodology and the results are interesting. I would recommend the following major revisions before the manuscript being considered for publication:*

General response: We sincerely appreciate your positive comments and kind recommendation on this work. We have synthesized new reference materials and performed additional characterizations to address the Reviewer's concerns and support our conclusion. Briefly, in response to the comments we have:

- Excluded Ostwald ripening process in this electrochemical environment by *in-situ* experiments.
- Captured the variation of an individual (NiCo)S_{1.33} particle under the applied voltage of 0.9 V.
- Observed the variation of (NiCo)S_{1.33} particle with reversed potentials following the initial amorphous shell formation.
- Corrected the errors in writing.

Specific responses to the comments and corresponding modifications are provided below.

Q1. *In Fig. 2 and also in the video clip, as the in-situ TEM experiment continues up to more than 30s, there are obvious total mass increase, particle coalition and particle agglomeration at the vicinity of the working electrode. Please specify the following:*

- a) Can you exclude Ostwald ripening process in this electrochemical environment? If so, how?*
- b) Instead of using a field of view with many (small) overlapped flakes, can you please choose one with less and clearly defined individual particles and present more convincing results?*

A1. Thank you for your thoughtful comments and important suggestion.

- a) After carefully considering your suggestion, we have designed two control experiments to exclude Ostwald ripening process in this electrochemical environment.

First of all, if it is Oswald ripening process that causes the increase of particle size and variation of morphology, the similar phenomenon should be observed under the same applied voltage conditions whether the particles are in contact with the working electrode or not. Therefore, we selected a nanoparticle, which has a comparable size with the central particle in Fig. 2e, without contacting the working electrode as the observation object and continuously monitor the morphology and size of this particle under the applied voltage of 0.9 V (Fig. R6 a). According to Fig. R6 b, it could be found that the morphology, size, and surface structure of this nanoparticle did not change during a period of up to 40 seconds.

Fig. R6. Real-time observation on the variation of a $(\text{NiCo})\text{S}_{1.33}$ particle away from the working electrode under the applied voltage of 0.9 V. **a**, The applied potential and corresponding current during the *in-situ* IT test. **b**, the HAADF and BF images of the $(\text{NiCo})\text{S}_{1.33}$ particle at different times.

In addition, according to the mechanism of Oswald ripening, the smaller particles will dissolve into the solvent, while the diameter of the large particles will continue to increase. Then, we could be able to observe the diameter reduction of the small particles during the dissolution process. Thus, we selected a

small particle with a diameter of 46.35 nm as the observation object and continuously monitor the morphology and size of this small particle under the applied voltage of 0.9 V (Fig. R7 a). As shown in Fig. R7 b, the diameter of this small particle increased to 71.23 nm within 30 seconds, which suggested that there is no dissolution process occurred in the electrochemical environment.

Fig. R7. Real-time observation on the variation of a small (NiCo)S_{1.33} particle on the working electrode under the applied voltage of 0.9 V. a, The applied potential and corresponding current during the *in-situ* IT test. **b,** the HAADF and BF images of the small (NiCo)S_{1.33} particle at different times.

Based on the above results, we can basically exclude the Ostwald ripening process in this electrochemical environment, which prove that the diameter increase and morphological variation of the (NiCo)S_{1.33} nanoparticles in Fig. 2e should be attributed to the surface reconstruction process. Please find that these parts have been add as Supplementary Fig. 29 and 30 in the revised Supplementary Information. And their corresponding discussions have been added at page 5-6 in the revised manuscript and page 33-34 in the revised supplementary information.

b) According to your suggestion, we selected an individual particle as the research object to observe the surface reconstruction process under the applied voltage of 0.9 V to provide a more convincing result. As shown in Fig. R8, this individual particle exhibited the similar behavior as those particles in Fig. 2d during

the surface reconstruction process. Please find that these parts have been add as **Supplementary Fig. 31** in the revised Supplementary Information. And their corresponding discussions have been added at page 6 in the revised manuscript and page 35 in the revised supplementary information.

Fig. R8. Real-time observation on the variation of an individual (NiCo)S_{1.33} particle on the working electrode under the applied voltage of 0.9 V. **a**, The applied potential and corresponding current during the *in-situ* IT test. **b**, the HAADF and BF images of the individual (NiCo)S_{1.33} particle at different times.

Q2. In Fig. 3, are the results including the HRTEM images in Fig. 3a truly collected in an *in-situ* environment with a flow of fresh KOH liquid and varied potentials constantly applied? If so, what's the estimated thickness of KOH in the liquid cell? How does KOH affect corresponding HRTEM, EELS and EDX results as shown in this figure?

A2. Thank you for your carefully reading and pointing this out.

The fixed spacer between Top and bottom chip is 500 nm. And due to the pressure difference between the inside and outside of the liquid cell, the silicon nitride film covered on the observation window will expand slightly after the sample holder entering the vacuum environment inside the electron microscope. The estimated thickness of KOH is about 550 to 600 nm.

The electrolyte has a scattering effect on the electron beam, resulting in a decrease in spatial and

energy resolution. Therefore, we cannot obtain atomic HAADF images and EELS spectrum with high energy resolution to accurately analyze the crystal structure and metal-oxygen bonds on the catalytic surface when the electrolyte exists. However, since the EDX technique is designed to collect the X-rays, the liquid layer has essentially no effect on the EDX results.

We used a strategy of draining the liquid to improve the spatial and energy resolution in order to eliminate the scattering effect of electrolyte (Fig. S27). For example, a specific voltage of 0.9 V (vs Pt RE) was applied for 21 seconds under electrolyte presence condition. Then, the electrolyte was emptied with the high purity argon, which can prevent the surface of catalyst particle from being oxidized, to improve the spatial resolution for structural and elemental analysis. After the collection of atomic HAADF images, EELS spectrum and EDX mapping, the liquid cell was refilled with electrolyte for subsequent IT test with another voltage and repeat this process until all voltage conditions are finished.

Q3. *Again in Fig. 3, Can you have a quantitative estimation of the relative atomic concentrations of O and S at 0.9V for 20S and does it match a lattice reconstruction consistent with the HRTEM result?*

A3. Thank you for your thoughtful comments and suggestion.

According to the atomic fraction of elements in $(\text{NiCo})\text{S}_{1.33}$ under the applied voltage of 0.9 V in Supplementary Table 2, the atomic fraction of O and S were 26.98 % and 30.72 %, respectively. The ratio of relative atomic concentration of O and S should be 0.88. However, the EDX data is a qualitative and semi-quantitative result. Strictly speaking, it can only reflect the relative content of different elements, which can be used to reveal the variation of elemental content, and it may not be suitable for giving an accurate quantitative estimation of the relative atomic concentrations.

Besides, the lattice sulfur atoms in the $(\text{NiCo})\text{S}_{1.33}$ surface are exchanged by oxygen atoms under the applied potential of 0.9 V, which further induces the formation of oxygen-sulphur coexisting surface in the range of 8.0 nm as $(\text{NiCo})\text{O}_x\text{S}_{1.33-x}$. It is worth noting that the oxygen-sulphur coexisting area has an uniform crystal structure as the $(\text{NiCo})\text{S}_{1.33}$ in bulk phase, which revealed that there is no lattice reconstruction occurred at 0.9 V. However, comparing the lattice fringes in the atomic HAADF images at OCV and 0.9 V (Fig. 3a) may give the illusion that the crystal structure has went through a reconstruction process. This is probably due to the slight rotation of the particles during process of draining and refilling the electrolyte in the liquid cell, which leads to the deviation of the crystal orientation. Thus, the HAADF images have shown the different lattice fringes of the spinel structure.

Q4. *How about a possible in-situ EC-TEM experiment and results with reversed potentials following the*

initial amorphous shell formation, or the structural evolution of relevant phases during I-V loop?

A4. Thank you for your thoughtful comments and important suggestion.

As shown in Fig. R9 a, we first applied a voltage of 0.9 V for 30 seconds and reversed the voltage to -0.9 V. It could be found that the diameter of (NiCo)S_{1.33} particle was increased from 189.91 nm to 236.51 nm during the application of positive voltage (Fig. R9 b). However, when the applied voltage was reversed to -0.9 V, the image contrast dropped sharply due to the effect of the electric field flip on the electron beam near the working electrode, and the variation of particle at this moment cannot be clearly observed. After the system automatically adjusts the image contrast, it can be observed that the particle has detached from the working electrode and eventually left the view field due to the electrolyte flow. The detachment of particle should be attributed to the charge accumulation on the catalyst surface during the application of positive voltage, which causes the particle surface to be negatively charged. And the negatively charged particle will attempt to repel with the working electrode when the applied voltage is reversed to -0.9 V. Combined with the current curve in Fig. R9 a, it could be found that the current also reversed in parallel with the applied voltage. However, the current tended to be zero as the particles were detached from the working electrode. In order to maintain the good sealing of the liquid cell, it is a pity that we could not add Nafion solution to fix the particles on the working electrode to implement your suggestion under the *in-situ* condition.

Fig. R9. Real-time observation on the variation of an individual (NiCo)S_{1.33} particle on the working electrode under the reversal process of applied voltage. **a**, The applied potential and corresponding current during the *in-situ* IT test. **b**, the HAADF and BF images of the individual (NiCo)S_{1.33} particle at different times.

Q5. *There are sporadic obvious typos and grammatical errors to be corrected. For example:*

a) line 31 "direction observation" => "direct observation"?

b) line 71 "...that driving..." => "...that drives..." or ...driving..."

Please proofread thoroughly.

A5. Thank you for your carefully reading and pointing these errors out. Please find that these errors have been corrected in the revised manuscript.

To Reviewer 3:

General comments: *In this manuscript the authors have investigated the surface structure evolution of Ni-Co-S of varied stoichiometry under conditions of OER. Recently transition metal chalcogenides have attracted significant attraction as electrocatalysts for various electrochemical processes including OER, ORR, and HER. Among these the performance of transition metal chalcogenides, especially, selenides and tellurides have shown tremendous promise for OER exhibiting some of the lowest overpotentials reported till date. In spite of their exceptional activity, there is still confusion regarding the actual catalytic species for these compositions. While traditionally it has been proposed that the chalcogenides are fully converted to (oxy)hydroxide surfaces under conditions of OER, there has been recent studies also which confirms that the chalcogenides actually form a mixed anionic (hydroxy)chalcogenide (selenide or telluride) surface, where the metal is coordinated to both Se/Te and OH.^[1,2] This manuscript constitutes an important step in that direction of gathering evidence for structural evolution of the surface under conditions of OER. However, the authors have only considered the least active (OER) and least stable chalcogenide, i.e. sulfides for their study. The inherent instability of the sulfides towards hydrolysis and even under ambient conditions dilutes the importance of these results, such that they cannot be extended to other highly active chalcogenides. What could have potentially been transformative for the field has become very system-specific without any far-reaching implications. This could have been important observations, however the narrow scope of these results along with other technical and scientific flaws makes it not ready for publication yet (in my opinion).*

General response: We sincerely appreciate your constructive comments and kind recommendation on this work. Your suggestions are very helpful for the design of our future research program and the development of our research project. We accept your suggestion and will conduct more in-depth *in-situ* studies on the electrochemical behavior of selenides and tellurides in our next research work to compensate for our limitation of focusing on sulphides. We have synthesized new reference materials and performed additional characterizations to address the Reviewer's concerns and support our conclusion. Briefly, in response to the comments we have:

- Verified the dominated sites for capturing OH* by DFT calculation.
- Verified that the (NiCo)S_{0.89}, (NiCo)S and (NiCo)S₂ are stoichiometric mixed metal sulphides by low magnification elemental mapping of Co and Ni.
- Recorded the current density under full exposure but zero bias.
- Excluded the electron beam irradiation effect on S leaching process by electron beam irradiation experiment, *in-situ* heating experiment and Thermogravimetric analysis (TGA) experiment.

Specific responses to the comments and corresponding modifications are provided below.

Q1. *Scrambling of the catalytic site: Ni vs Co*

The authors have proposed that Co is the actual catalyst site in the most active composition (NiCo₂Se₄) although Ni assists to boost performance. However, from the DFT calculation it was shown that the Ni d-band center was higher and closer to the Fermi surface in the band structure diagram. This would imply -OH binding to the Ni-site (d-orbitals) as opposed to Co, making Ni as the catalytic site. This is also supported by the observation that Ni- pre oxidation peak precedes the OER onset, which suggests that Ni pre-oxidation constitutes catalyst activation. If Co was the actual catalytic site, OER onset will not follow the Ni-site activation step.

In fact in the mixed metal chalcogenides, understanding of scrambling of the catalyst sites is a major challenge. The authors may utilize DFT studies to actually investigate the site activation step between Co and Ni and correlate that with the observed (bulk) overpotential to offer better insight into the actual catalyst site.

A1. Thank you for pointing this out.

The Co and Ni pre-oxidation peaks of (NiCo)S_{1.33} were observed at 1.21 and 1.26 V, respectively (Fig. S24 a). And the Co and Ni pre-oxidation peaks both precede the OER onset, which suggests that they jointly constitute the catalyst activation. However, it is difficult to precisely determine the dominated active site just by the location of the Co and Ni pre-oxidation peaks and their relative position. Thus, we

have calculated the d-band center of Co and Ni in (NiCo)S_{1.33} and monitored the characteristic peaks of M-OH by operando Raman spectra to comprehensively investigate their adsorption capacity for -OH, which could help us to identify the dominated catalyst site through a combination of theoretical and experimental results.

Firstly, as shown in Fig. S19 and Table S1, the d-band center of Co and Ni in (NiCo)S_{1.33} were calculated to be -1.1185 and -1.4842, respectively. This result reveals that the d-band center of Co in (NiCo)S_{1.33} was higher and closer to the Fermi surface than that of Ni, which implied -OH binding to the Co-site (d-orbitals) as opposed to Ni and making Co as the dominant catalytic site. Furthermore, the d-band center of Co and Ni in O-substituted (NiCo)S_{1.33} were calculated to be -1.1720 and -1.3766 (Figure S37 and Table S6), respectively. The comparison of the metal d-band center in the pure sulphides and corresponding lattice O-substituted sulphides in Fig. 4d has revealed the upper shift of Co and Ni d-band center in (NiCo)S_{1.33} due to the oxygen-sulphur substitution process. And the ΔE_d of Co and Ni were calculated to be 0.0130 and 0.1076, respectively. This result suggests that the upper shift of Ni d-band center to the Fermi surface is more prominent than that of Co, which implies an enhancement of the adsorption capacity of Ni for the reaction intermediates. But Co is still playing the dominated active site for adsorbing the reaction intermediates after the oxygen-sulphur substitution process.

Secondly, the characteristic peaks of M-OH in (NiCo)S_{1.33}, which were extracted from the operando Raman spectra (Fig. S45), should mainly come from the contribution of Co-OH due to the great gap between the Co-OOH and Ni-OOH, indicating that Co is suggested to be the dominant metal active site for capturing OH* and promote the occurrence of OER reaction, while Ni plays an auxiliary role.

Follow your suggestion, we have utilized the DFT studies to calculate the adsorption energy of OH* on catalytically active Co and Ni sites respectively. As shown in Fig. R5, the adsorption energy of OH* on Co sites on (110) terminated surfaces of (NiCo)S_{1.33} and O-substituted (NiCo)S_{1.33} were calculated to be -2.02 and -2.11, respectively. In comparison, the adsorption energy of OH* on Ni sites of both (NiCo)S_{1.33} and O-substituted (NiCo)S_{1.33} were lower than that on Co sites, which suggested that Co is the dominated active sites. Besides, the adsorption energy of OH* on Ni sites was revealed to lift more obvious than that of Co, which implied the enhancement of OH* adsorption capacity of Ni.

Combining the results of DFT calculation and Raman spectra, we suggested that Co is the dominated active site. Please find that these parts have been add as **Supplementary Fig. 56** in the revised supplementary information. And their corresponding discussions have been added at page 9 in the revised manuscript.

Q2. *Except for NiCo₂Se₄ the other 3 compositions reported in this manuscript are mixtures of binary sulfides of Ni-S and Co-S. It is not advisable to compare activity of a stoichiometric mixed metal selenides with mixture of binaries, especially when both the binaries (Ni-S and Co-S) are active for OER to varied extent. Mixture of binaries will expectedly lead to performance which is either representative of the average of the individual components (Ni-S and Co-S phases) or the dominant one.*

A2. Thank you for pointing this out.

Firstly, as shown in Fig. 1a and Fig. S2, the diffraction peak of the (NiCo)S_{0.89} is consistent with the standard peak of Co₉S₈ (Fig. S2 a) with the peak position shifting to a lower angle. And the characteristic peaks of (NiCo)S and (NiCo)S₂ are both in the middle of the two monometallic sulfides. Besides, the scanning rate was set to 0.02 °/s for acquiring an adequate resolution of XRD Spectrum. If their compositions are mixtures of binary sulfides of Ni-S and Co-S, the diffraction peaks should also be mixtures of two different characteristic peaks. For example, the XRD pattern of binary sulphides (NiS and CoS) will exhibit the mixed diffraction peaks of NiS and CoS, which is obviously different from the XRD patterns of (NiCo)S reported in our manuscript. This kind of a peak shift could be attributed to the different ionic radii of the Ni and Co cations in the same crystal cell, which causes the changes of lattice parameters and suggests a solid solution property.

Secondly, the atomic elemental distribution characterization of (NiCo)S_{0.89}, (NiCo)S and (NiCo)S₂ revealed that the Co and Ni atoms inside each structure have a similar coordination form with S atoms due to the same occupation of two metal atoms (Fig. S8-10), which proved that each atomic column has both cobalt and nickel atoms in it. Besides, the Co and Ni are evenly distributed in each particle of (NiCo)S_{0.89}, (NiCo)S and (NiCo)S₂ (Fig. R10). But for mixture of binaries, the elemental distribution of Co and Ni should exhibit a clear distinction because each single particle has only one kind of metallic element.

Fig. R10. Elemental mapping of Co and Ni of (NiCo)S_{0.89}, (NiCo)S and (NiCo)S₂.

Based on above results, it could be confirmed that the (NiCo)S_{0.89}, (NiCo)S and (NiCo)S₂ reported in our manuscript are stoichiometric mixed metal sulphides. Please find that these parts have been add as **Supplementary Fig. 11** in the revised supplementary information. And their corresponding discussions have been added at page 4 in the revised manuscript and page 15 in the revised supplementary information.

Q3. The Co pre-oxidation peak is typically observed at ~1.1 v vs RHE. However, this peak was significantly in the LSV plots of even the bulk samples. Does that signify that Co-site is not getting oxidized under these conditions. If so, then Co is not the actual catalytic site and the premise of discussion in the manuscript is wrong.

A3. Thank you for pointing this out.

The Co pre-oxidation peak of (NiCo)S_{1.33} was observed at 1.21 V, which proved that Co-site has been oxidized. In fact, the Co pre-oxidation peak of different catalysts could locate at various potential. For example, the Co pre-oxidation peak of iron-cobalt oxide was observed at ~1.2 V vs RHE (Nat. Commun. 2017, 8, 2022), while the Co pre-oxidation peak of nickel-cobalt sulphide was observed at ~1.2 V vs RHE (Adv. Funct. Mater. 2016, 26, 4661). In this manuscript, we suggested that Co is the dominant catalytic site and Ni plays a supporting role, which could be supported by the oxidization peak of Co in CV curves, DFT results and Raman spectra.

Q4. Although the premise of in situ electrochemical studies on microscopic sample space is intriguing, I

have a major concern regarding the effect of the accelerated e-beam (from TEM) on the observed electrochemical behavior. The e-beam will lead to surface impingement of large density of electrons that will be dissipated throughout the sample at various rates depending on the intergrain conductivity. NiCo₂Se₄ with higher conductivity will dissipate the electrons faster leading to larger current density. However, that current density is not necessarily from OER but rather from facile electron transfer across the catalyst surface. The binary sulfides on the other hand, will lead to sluggish growth of current density since the large density of impinged electrons are trapped on the surface. The authors should perform more control experiments including the one where the current density is recorded under full exposure but zero bias.

A4. Thank you for your thoughtful comments and important suggestion.

According to your suggestion, we perform a control experiment to record the variation of current density with and without electron beam irradiation under the applied voltage of 0 V. As shown in Fig. R11, it could be found that there is no significantly fluctuation of current density when the electron beam switched from the off state to on state. And the current density did not fluctuate significantly during the 5 minutes of electron beam irradiation. Besides, the Fig. S25 has shown the in-situ CV curves of (NiCo)S_{1.33} separately under the two conditions of electron beam isolation and continuous electron beam irradiation, which suggested that the effect of electron beam irradiation is negligible. Please find that these parts have been add as Supplementary Fig. 27 in the revised Supplementary Information. And their corresponding discussions have been added at page 5 in the revised manuscript ad page 31 in the revised supplementary information.

Fig. R11. Real-time record on the variation of current density with and without electron beam irradiation under the applied voltage of 0 V.

Q5. The other concern regarding accelerated e-beam exposure is local surface heating where it has been observed previously that local temperature of the beam spot can go as high as several hundred of degrees. Sulfides are known to degrade by loss of S atoms at elevated temperatures. In fact, prolonged TEM imaging of sulfides under normal conditions (high vacuum and no reactive atmosphere exposure) also lead to such decomposition and loss of S. Hence, it might be wrong to assume that the S loss observed in these TEM imaging is due to progression of OER. The authors should also study S leaching (if any) from the bulk samples through analytical techniques, quantify that and then correlate with the electron microscopy studies.

A5. Thank you for your thoughtful comments and important suggestion.

According to your suggestion, we perform three different control experiments to answer your question about S leaching process, which are electron beam irradiation experiment, *in-situ* heating experiment and Thermogravimetric analysis (TGA) experiment.

Experiment No.1 Electron beam irradiation experiment

If the electron beam irradiation will cause the leaching of S, the prolonged irradiation will inevitably lead to the reduction of sulfur content in the sample. Thus, we have designed an irradiation experiment to investigate whether electron beam irradiation leads to the leaching of sulphur.

First of all, the screen current is an important parameter to measure the irradiation intensity of electron beam. The higher screen current, the more electrons will bombard the sample per unit time. In order to minimize the influence of electron beam irradiation on the sample, the screen current was set to 30 pA during the *in-situ* electrochemical-TEM experiment.

Based on the parameter setting of screen current in the *in-situ* EC-TEM experiment, the $(\text{NiCo})\text{S}_{1.33}$ particles were continuously irradiated for 10 minutes at the screen current of 30 pA and 300 pA respectively in the irradiation experiment. And the elemental distribution and relative content of S before and after irradiation were characterized and measured. In addition, considering the difference between vacuum and liquid environments, we have conducted the irradiation experiments in vacuum and liquid (0.1 M KOH) environment respectively to obtain more comprehensive comparison results.

Fig. R12. Electron beam irradiation experiment of $(\text{NiCo})\text{S}_{1.33}$ particles at the screen current of 30 pA in the vacuum environment. **a**, Low magnification HAADF image and elemental mapping of S before irradiation. **b**, High magnification HAADF image and elemental mapping of S before irradiation. **c**, Screenshot of the screen current parameter before irradiation. **d**, Relative contents of different elements and corresponding EDX element spectra before irradiation. **e**, Low magnification HAADF image and elemental mapping of S after irradiation. **f**, High magnification HAADF image and elemental mapping of S after irradiation. **g**, Screenshot of the screen current parameter after irradiation. **h**, Relative contents of different elements and corresponding EDX element spectra after irradiation.

As shown in Fig. R12, the relative content and distribution of S did not change significantly after 10 minutes irradiation at the screen current of 30 pA. This result revealed that the electron beam irradiation with the screen current of 30 pA did not lead to the leaching of sulphur in the vacuum environment.

Fig. R13. Electron beam irradiation experiment of $(\text{NiCo})\text{S}_{1.33}$ particles at the screen current of 30 pA in the liquid environment. **a**, Low magnification HAADF image and elemental mapping of S before irradiation. **b**, High magnification HAADF image and elemental mapping of S before irradiation. **c**, Screenshot of the screen current parameter before irradiation. **d**, Relative contents of different elements and corresponding EDX element spectra before irradiation. **e**, Low magnification HAADF image and elemental mapping of S after irradiation. **f**, High magnification HAADF image and elemental mapping of S after irradiation. **g**, Screenshot of the screen current parameter after irradiation. **h**, Relative contents of different elements and corresponding EDX element spectra after irradiation.

As shown in Fig. R13, the relative content and distribution of S did not change significantly after 10 minutes irradiation at the screen current of 30 pA. This result revealed that the electron beam irradiation with the screen current of 30 pA did not lead to the leaching of sulphur in the liquid environment. (The screen current reduced to 25 pA due to the scattering effect of liquid on electron beam. The extra Si signal in the EDX spectrum comes from the silicon nitride film of the electrochemical chip.)

Fig. R14. Electron beam irradiation experiment of $(\text{NiCo})\text{S}_{1.33}$ particles at the screen current of 300 pA in the vacuum environment. **a**, Low magnification HAADF image and elemental mapping of S before irradiation. **b**, High magnification HAADF image and elemental mapping of S before irradiation. **c**, Screenshot of the screen current parameter before irradiation. **d**, Relative contents of different elements and corresponding EDX element spectra before irradiation. **e**, Low magnification HAADF image and elemental mapping of S after irradiation. **f**, High magnification HAADF image and elemental mapping of S after irradiation. **g**, Screenshot of the screen current parameter after irradiation. **h**, Relative contents of different elements and corresponding EDX element spectra after irradiation.

As shown in Fig. R14, the relative content and distribution of S did not change significantly after 10 minutes irradiation at the screen current of 300 pA. This result revealed that the electron beam irradiation with the screen current of 300 pA did not lead to the leaching of sulphur in the vacuum environment.

Fig. R15. Electron beam irradiation experiment of $(\text{NiCo})\text{S}_{1.33}$ particles at the screen current of 300 pA in the liquid environment. a, Low magnification HAADF image and elemental mapping of S before irradiation. b, High magnification HAADF image and elemental mapping of S before irradiation. c, Screenshot of the screen current parameter before irradiation. d, Relative contents of different elements and corresponding EDX element spectra before irradiation. e, Low magnification HAADF image and elemental mapping of S after irradiation. f, High magnification HAADF image and elemental mapping of S after irradiation. g, Screenshot of the screen current parameter after irradiation. h, Relative contents of different elements and corresponding EDX element spectra after irradiation.

As shown in Fig. R15, the relative content and distribution of S did not change significantly after 10 minutes irradiation at the screen current of 300 pA. This result revealed that the electron beam irradiation with the screen current of 300 pA did not lead to the leaching of sulphur in the liquid environment.

The above experimental results have revealed that the continuous electron beam irradiation is not the cause of sulfur leaching in our *in-situ* EC TEM at the current of 30 pA.

Experiment No.2 *in-situ* heating experiment

In order to investigate whether the high temperature will lead to the leaching of sulphur in $(\text{NiCo})\text{S}_{1.33}$, we have designed an *in-situ* heating experiment to monitor the variation of relative content and distribution of S under the temperature of $500\text{ }^{\circ}\text{C}$.

As shown in Fig. R16, the temperature raised up from room temperature ($25\text{ }^{\circ}\text{C}$) to $500\text{ }^{\circ}\text{C}$ within 950 seconds and held at $500\text{ }^{\circ}\text{C}$ for 10 minutes. The relative content and distribution of S was obtained at $25\text{ }^{\circ}\text{C}$ and $500\text{ }^{\circ}\text{C}$ respectively to monitor the changes before and after heating. Besides, this heating experiment was conducted in a vacuum environment.

Fig. R16. Screenshot of the temperature and time parameters of *in-situ* heating experiment from the control software.

Fig. R17. *in-situ* heating experiment of (NiCo)S_{1.33} particles at the screen current of 30 pA in the vacuum environment. a, the temperature-time curve. **b**, Low magnification HAADF image and elemental mapping of S at 25 °C. **c**, High magnification HAADF image and elemental mapping of S at 25 °C. **d**, Screenshot of the screen current parameter at 25 °C. **e**, Relative contents of different elements and corresponding EDX element spectra at 25 °C. **f**, Low magnification HAADF image and elemental mapping of S at 500 °C. **g**, High magnification HAADF image and elemental mapping of S at 500 °C. **h**, Screenshot of the screen current parameter at 500 °C. **i**, Relative contents of different elements and corresponding EDX element spectra at 500 °C.

As shown in Fig. R17, the relative content and distribution of S did not change significantly after heating up to 500 °C at the screen measured current of 30 pA. This result revealed that the high temperature is not the cause of sulfur leaching in our *in-situ* EC-TEM at the current of 30 pA. In addition, the electrolyte with room temperature was constantly flowing during the *in-situ* EC TEM experiment, which would counteract the heating effect caused by electron beam irradiation at a certain extent and keep the catalyst particles in a constant temperature state.

Experiment No.3 Thermogravimetric analysis (TGA) experiment

In order to investigate whether the high temperature will lead to the leaching of sulphur in $(\text{NiCo})\text{S}_{1.33}$ through another analytical technique and then correlate with the electron microscopy studies. We have performed the thermogravimetric analysis to carry out a macroscopical result to make up for the deficiency of microscopic electron microscope analysis.

As shown in Fig. R18, the overall quality of $(\text{NiCo})\text{S}_{1.33}$ powder did not change significantly during the process of heating up to 500 °C in the nitrogen environment.

Fig. R18. Thermogravimetric analysis of $(\text{NiCo})\text{S}_{1.33}$ powder during the process of heating up to 500 °C in the N_2 environment.

According to the results of above three comparative experiments, it could be confirmed that the leaching of sulphur observed in the *in-situ* EC-TEM experiment is due to progression of electrochemical reaction. Please find that these parts have been add as **Supplementary Fig. 37-43** in the revised Supplementary Information. And their corresponding discussions have been added at page 7 in the revised manuscript and page 41-48 in the revised supplementary information.

In addition to the revisions made above, we proof-read the whole manuscript and made some necessary minor revisions to the writing. Thank you for your time and consideration.

Sincerely,
Pinxian Xi

REVIEWERS' COMMENTS

Reviewer #1 (Remarks to the Author):

The authors have addressed the reviewer's comments satisfactorily. Therefore, the manuscript can be accepted for publication.

Regarding the comments from reviewer#3, I went through the response from the authors. The authors have provided substantial additional experimental results together with DFT calculations to address the concerns raised by reviewer#3. The response has addressed the concerns evidently, in my opinion. Therefore, I recommend accepting the manuscript for publication.

Reviewer #2 (Remarks to the Author):

Overall, I'm happy with the revision and would recommend to publish the revised version with any editorial corrections as needed.

Pinxian Xi

State Key Laboratory of Applied Organic Chemistry
College of Chemistry and Chemical Engineering
Lanzhou University
Lanzhou 730000, China.
E-mail: xipx@lzu.edu.cn

Mar. 20th, 2023

Manuscript ID NCOMMS-22-39322A

Dear Reviewers,

Thank you for the positive comments on our revised paper “*Revealing the Lattice Sulphur-Oxygen Substitution Pathway of Metal Sulphide Electrocatalysts at Pre-catalytic Stage during Oxygen Evolution Reaction*” (Manuscript ID NCOMMS-22-39322A). We appreciate your efforts and time in reviewing our manuscript. Our answers to Reviewers’ comments are given below:

To Reviewer 1:

Comments: The authors have addressed the reviewer's comments satisfactorily. Therefore, the manuscript can be accepted for publication.

Response: Thank you for your recognition of this work. We sincerely appreciate your efforts and time in reviewing our manuscript.

To Reviewer 2:

Comments: Overall, I'm happy with the revision and would recommend to publish the revised version with any editorial corrections as needed.

Response: Thank you for your recognition of this work. We sincerely appreciate your efforts and time in

reviewing our manuscript.

To Reviewer 3:

Comments: Regarding the comments from reviewer#3, I went through the response from the authors. The authors have provided substantial additional experimental results together with DFT calculations to address the concerns raised by reviewer#3. The response has addressed the concerns evidently, in my opinion. Therefore, I recommend accepting the manuscript for publication.

Response: Thank you for your recognition of this work. We sincerely appreciate your efforts and time in reviewing our manuscript.

In addition, we proof-read the entire manuscript and made the necessary changes to the text and figures as requested by the editor. Thank you for your time and consideration.

Sincerely,
Pinxian Xi